# Merlin[S13] phosphorylation regulates meningioma Wnt signaling and magnetic resonance imaging features

Charlotte D. Eaton[1,2,3], Lauro Avalos[2,4], S. John Liu [1,2,3], Zhenhong Chen[1,2,3], Naomi Zakimi[1,2,3], Tim Casey-Clyde [1,2,3], Paola Bisignano [5], Calixto-Hope G. Lucas [6], Erica Stevenson[7,8], Abrar Choudhury [1,2,3], Harish N. Vasudevan [1,2], Stephen T. Magill[9], Jacob S. Young[1,2,3], Nevan J. Krogan [7,8], Javier E. Villanueva-Meyer[2,4], Danielle L. Swaney [7,8] & David R. Raleigh [1,2,3] ✉

Meningiomas are associated with inactivation of *NF2*/Merlin, but approximately one-third of meningiomas with favorable clinical outcomes retain Merlin expression. Biochemical mechanisms underlying Merlin-intact meningioma growth are incompletely understood, and non-invasive biomarkers that may be used to guide treatment de-escalation or imaging surveillance are lacking. Here, we use single-cell RNA sequencing, proximity-labeling proteomic mass spectrometry, mechanistic and functional approaches, and magnetic resonance imaging (MRI) across meningioma xenografts and patients to define biochemical mechanisms and an imaging biomarker that underlie Merlin-intact meningiomas. We find Merlin serine 13 (S13) dephosphorylation drives meningioma Wnt signaling and tumor growth by attenuating inhibitory interactions with β-catenin and activating the Wnt pathway. MRI analyses show Merlin-intact meningiomas with S13 phosphorylation and favorable clinical outcomes are associated with high apparent diffusion coefficient (ADC). These results define mechanisms underlying a potential imaging biomarker that could be used to guide treatment de-escalation or imaging surveillance for patients with Merlin-intact meningiomas.

Meningiomas arising from the meningothelial lining of the central nervous system comprise over 40% of primary intracranial tumors[1,2], and ~1% of humans will develop a meningioma in their lifetime[3]. Meningiomas are treated with surgery and radiotherapy, and systemic therapies remain ineffective or experimental for patients with meningiomas[4]. Bioinformatic investigations have revealed biological drivers and therapeutic vulnerabilities underlying meningiomas with unfavorable outcomes[5–13], and clinical trials of new therapeutic strategies to treat patients with meningiomas that are resistant to standard interventions are underway[4]. Nevertheless, most meningiomas are benign and some meningiomas can be safely observed with serial magnetic resonance imaging (MRI)[14]. Imaging surveillance can spare

[1]Department of Radiation Oncology, University of California San Francisco, San Francisco, CA, USA. [2]Department of Neurological Surgery, University of California San Francisco, San Francisco, CA, USA. [3]Department of Pathology, University of California San Francisco, San Francisco, CA, USA. [4]Department of Radiology and Biomedical Imaging, University of California San Francisco, San Francisco, CA, USA. [5]Department of Molecular Physiology and Biophysics, Vanderbilt University, Nashville, TN, USA. [6]Department of Pathology, Johns Hopkins University, Baltimore, MD, USA. [7]J. David Gladstone Institutes, California Institute for Quantitative Biosciences, San Francisco, CA, USA. [8]Department of Cellular and Molecular Pharmacology, University of California San Francisco, San Francisco, CA, USA. [9]Department of Neurological Surgery, Northwestern University, Chicago, IL, USA. ✉e-mail: david.raleigh@ucsf.edu

patients from morbidities associated with potentially unnecessary surgical or radiotherapy treatments, but current meningioma classification systems rely on histological and molecular analyses of tumor tissue[15]. Thus, there is an unmet need to develop non-invasive imaging biomarkers that could be used to predict meningioma outcomes or could be used to guide treatment de-escalation or surveillance for the most common primary intracranial tumor.

Meningiomas are often associated with inactivation of the tumor suppressor *NF2*/Merlin[16], but approximately one-third of meningiomas are Merlin-intact and have favorable clinical outcomes[7–9,11,17,18]. Merlin is a member of the FERM family of proteins (Four-point-one, Ezrin, Radixin, and Moesin) that link the cytoskeleton to the plasma membrane. These proteins are comprised of a FERM domain, an α-helical domain, and a C-terminal domain (CTD). Merlin-intact meningiomas can encode somatic short variants (SSVs) targeting *TRAF7*, *PIK3CA*, *AKT1*, *KLF4*, or the Hedgehog pathway[17,18], but some of these variants do not drive meningioma tumorigenesis in mice, and others do not drive meningioma cell proliferation or susceptibility to molecular therapy in vitro[19–21]. These data suggest that some SSVs in Merlin-intact meningiomas may be passengers that do not influence meningioma tumorigenesis or perhaps modifiers (rather than drivers) meningioma biology. More broadly, these data indicate that biochemical

mechanisms underlying Merlin-intact meningiomas are incompletely understood.

Here, we test the hypothesis that understanding signaling mechanisms associated with *NF2*/Merlin itself may shed light on meningioma biology and elucidate strategies to define meningioma biology pre-operatively using non-invasive imaging techniques.

## Results

To study Merlin signaling mechanisms in preclinical meningioma models, CH-157MN human meningioma cells lacking endogenous Merlin[22] were transduced with a doxycycline-inducible *NF2* construct and grown as xenografts in mice (Fig. 1a). Merlin rescue with doxycycline in CH-157MN xenografts did not influence meningioma histology, growth, or overall survival compared to xenografts in mice without doxycycline (Fig. 1b, c and Supplementary Fig. 1). To validate these findings, IOMM-Lee human meningioma cells encoding endogenous Merlin[23] were transduced with short-hairpin RNAs (shRNAs) suppressing *NF2* (sh*NF2*), or non-targeted control shRNAs (shNTC), and grown as xenografts in mice (Supplementary Fig. 2a). Merlin loss in IOMM-Lee xenografts did not influence meningioma histology, growth, or overall survival compared to mice with IOMM-Lee xenografts expressing shNTC (Supplementary Fig. 2b-e).

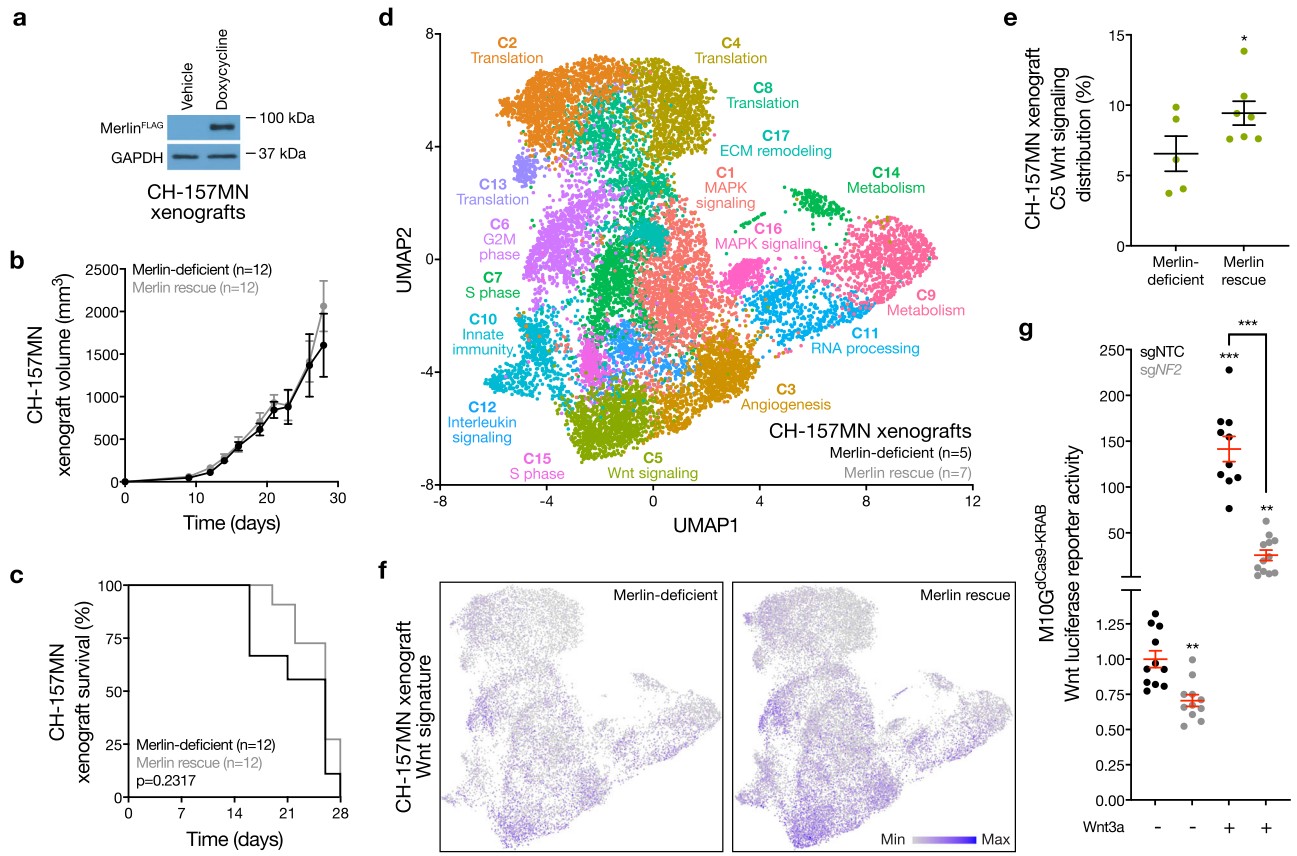

**Fig. 1 | Merlin drives meningioma Wnt signaling. a** Immunoblots for FLAG (Merlin) or GAPDH in CH-157MN meningioma xenografts with or without 24 h of doxycycline-inducible Merlin rescue (200 μg/ml). Representative of four experimental repeats. **b** CH-157MN xenograft measurements in NU/NU mice with (*n* = 12) or without (*n* = 12) doxycycline-inducible Merlin rescue as in (**a**). **c** Kaplan-Meier survival curve for CH-157MN xenograft overall survival in NU/NU mice as in (**b**) (log-rank test). **d** Uniform manifold approximation and projection (UMAP) of single-cell RNA sequencing transcriptomes of 40,765 CH-157 cells from 12 xenografts (*n* = 5 Merlin-deficient CH-157MN xenografts, *n* = 7 CH-157MN xenografts with Merlin rescue) as in (**a–c**) colored by assignments from Louvain clustering.

**e** Quantification of single-cell types from (**d**) that were differentially enriched in Merlin-deficient compared to Merlin rescue xenografts. **f** Feature plots for the MSigDB Hallmark Wnt target gene expression signature in single-cell transcriptomes from Merlin-deficient (*n* = 5) compared to Merlin rescue xenografts (*n* = 7). **g** TOP-Flash Tcf/Lef luciferase reporter assay in M10G$^{dCas9-KRAB}$ meningioma cells expressing non-targeted control sgRNAs (sgNTC) or sgRNAs suppressing *NF2* (sg*NF2*) with or without 24 h of Wnt3a treatment (100 ng/μl). Representative of three experimental repeats. Lines represent means, and error bars represent standard error of the means. *$P \le 0.05$, **$P \le 0.01$, ***$P \le 0.0001$ (Student's *t* test, one sided). Source data are provided as a Source Data file.

To shed light on the gene expression programs underlying these unexpected phenotypes, single-cell RNA sequencing was performed on 40,765 human meningioma cells from 5 Merlin-deficient CH-157MN xenografts and 7 CH-157MN xenografts with Merlin rescue (Supplementary Fig. 3a). Uniform manifold approximation projection (UMAP) and Louvain clustering defined 17 meningioma xenograft cell states (Fig. 1d), which were distinguished using differentially expressed genes and phases of the cell cycle (Supplementary Fig. 3b, c and Supplementary Data 1). Comparison of meningioma cell states across xenografts revealed that only one cell cluster was enriched in Merlin rescue xenografts compared to Merlin-deficient xenografts (Fig. 1e). This cluster was distinguished by expression of the Wnt pathway effector *CTNNB1*/β-catenin (Supplementary Fig. 3b). In support of these data, Wnt target genes were also enriched in single cells from Merlin rescue xenografts compared to Merlin-deficient xenografts (Fig. 1f).

Markers of Wnt pathway activation have been identified in human meningiomas[24,25], but how Wnt signals are transduced through meningioma cells is unknown. Canonical Wnt signaling proceeds through β-catenin, which is degraded by a constitutively active destruction complex in the absence of Wnt stimulation[26]. Wnt stimulation induces PP1A to de-phosphorylate and inactivate the β-catenin destruction complex[27], allowing β-catenin to localize to the plasma membrane or to the nucleus in complex with Tcf/Lef, a transcriptional co-activator that induces the expression of Wnt target genes.

To determine if Merlin regulates Wnt signaling in meningioma cells, non-malignant M10G human meningioma cells, which encode *NF2*/Merlin[28], were engineered to stably express CRISPR interference (CRISPRi) machinery (dCas9-KRAB)[29] and transduced with non-targeted control sgRNAs (sgNTC) or sgRNAs suppressing endogenous *NF2* (sg*NF2*) (Supplementary Fig. 4a). Co-transfection of the TOP-Flash Tcf/Lef Wnt luciferase reporter and treatment with recombinant Wnt3a or vehicle control revealed Merlin suppression attenuated Wnt signaling in meningioma cells (Fig. 1g). *NF2*/Merlin inactivation is also associated with schwannoma tumorigenesis[16], and Merlin suppression with sg*NF2* in human HEI-193 schwannoma cells[30] stably expressing CRISPRi machinery also attenuated Wnt signaling compared to sgNTC (Supplementary Fig. 4b, c).

To identify candidates mediating Merlin regulation of the Wnt pathway, M10G cells were transduced with doxycycline-inducible wildtype or cancer-associated missense Merlin constructs (L46R, A211D) that encoded C-terminal FLAG and APEX2 tags to enable subcellular localization experiments and proximity labeling proteomic mass spectrometry interrogation of candidate interactors[31]. Immunofluorescence or immunoblots after biochemical subcellular fractionation demonstrated decreased stability and re-localization of Merlin^L46R and Merlin^A211D compared to Merlin^WT that was also evident after streptavidin labeling of biotinylated peptides in proximity to Merlin constructs (Fig. 2a and Supplementary Fig. 5a). Streptavidin pulldown and proximity-labeling proteomic mass spectrometry identified β-catenin adjacent to Merlin^WT but not Merlin^L46R, and also revealed that β-catenin was diminished in proximity to Merlin^A211D (Fig. 2b and Supplementary Data 2). In support of these data, Merlin^L46R and Merlin^A211D were unable to rescue Wnt signaling in M10G cells with CRISPRi suppression of endogenous *NF2* (Fig. 2c). β-catenin was not degraded by loss of *NF2* in meningioma cells (Supplementary Fig. 5b), and subcellular fractionation of meningioma cells after doxycycline-inducible Merlin rescue showed β-catenin was distributed across the plasma membrane, cytoplasm, cytoskeleton, and nucleus with Merlin^L46R and Merlin^A211D rescue, but was only enriched at the plasma membrane and in the nucleus with Merlin^WT rescue (Fig. 2d). β-catenin suppression using siRNAs (si*CTNNB1*) inhibited meningioma Wnt signaling (Fig. 2e and Supplementary Fig. 5c), but Merlin was required for maximal Wnt pathway activation in meningioma cells even after β-catenin overexpression (Fig. 2f and Supplementary Fig. 5d). In the absence of Wnt3a, Merlin overexpression did not activate the Wnt pathway in

meningioma cells with or without endogenous Merlin, but endogenous Merlin suppression attenuated Wnt signaling (Fig. 2g, h). In the presence of Wnt3a, overexpression of Merlin activated the Wnt pathway in meningioma cells regardless of endogenous Merlin status (Fig. 2g, h). These data suggest Merlin regulates the Wnt pathway through a feed-forward mechanism that influences Merlin/β-catenin interaction and β-catenin subcellular localization.

Visual inspection of an available crystal structure of Merlin (PDB 4ZRJ) revealed that the L46R and A211D cancer-associated missense mutations were located on α-helices embedded in hydrophobic pockets of the Merlin FERM domain (Fig. 3a, b). These data suggest that charged amino acid substitutions of L46 or A211 may destabilize the secondary structure of the protein. The Merlin N-terminal domain (NTD) is unique among FERM family members, and structural modeling showed the NTD is a flexible, 19-residue α-helix that protrudes from the surface of the protein (Fig. 3a). Overexpression of Moesin, or Merlin rescue using a construct lacking the NTD (Merlin^ΔNTD), was unable to rescue Wnt signaling in Merlin-deficient meningioma cells (Fig. 3c and Supplementary Fig. 6a). Thus, we hypothesized the Merlin NTD may regulate Merlin/β-catenin interaction and Wnt signaling in meningiomas.

Sequence analysis of the Merlin NTD showed a phosphorylation site on serine 13 (S13) within consensus motifs for a kinase (PKC) and a phosphatase (PP1A) that are core components of the Wnt pathway and were also identified in proximity to Merlin^FLAG-APEX2 constructs in meningioma cells (Supplementary Data 2). Proteomic proximity-labeling mass spectrometry of Merlin constructs with unphosphorylatable (S13A) or phospho-mimetic (S13D) substitutions revealed β-catenin adjacent to Merlin^S13D but not Merlin^S13A (Fig. 2a, b). Immunoprecipitation and immunoblots validated β-catenin interaction with Merlin^S13D but not Merlin^S13A in meningioma cells (Fig. 3d), but in vitro recombinant protein binding assays showed that the Merlin NTD (irrespective of unphosphorylatable or phospho-mimetic substitutions at S13) was not sufficient for interaction with β-catenin (Supplementary Fig. 6b). These data suggest that the tertiary structure of Merlin (or additional scaffolding proteins) may be necessary for interaction with β-catenin. Rescue of Merlin^S13D but not Merlin^S13A sequestered β-catenin at the plasma membrane in meningioma cells without causing dramatic differences in the subcellular localization of Merlin itself (Fig. 2d), and Merlin^S13A but not Merlin^S13D rescued Wnt signaling in meningioma cells lacking *NF2* (Fig. 3e and Supplementary Fig. 6c). Merlin^S13D rescue attenuated meningioma cell proliferation in vitro (Fig. 3f), inhibited meningioma xenograft growth in vivo (Fig. 3g, Supplementary Fig. 6d-g, and Supplementary Data 3), and prolonged overall survival compared to mice with Merlin^WT or Merlin^S13A rescue (Fig. 3h). Immunoblots of meningioma cell lysates after shRNA suppression of PKC ± phosphatase inhibition revealed a Merlin doublet that was eliminated with suppression of PKCα or PKCγ, or with overexpression of Merlin^S13A (Supplementary Fig. 6h). Generation of a phospho-specific antibody recognizing Merlin^pS13 (Supplementary Fig. 6i) showed small-interfering RNAs (siRNAs) suppressing PP1A increased Merlin^pS13 immunoblot intensity but siRNAs suppressing PKCα and PKCγ decreased Merlin^pS13 immunoblot intensity compared to non-targeted control siRNAs (siNTC) in meningioma cells (Fig. 3i, j). Moreover, Merlin-dependent Wnt signaling was attenuated by siRNAs suppressing PP1A and activated by siRNAs suppressing PKCγ in meningioma cells compared to siNTC (Fig. 3k and Supplementary Fig. 6j). These data suggest that PKC and PP1A regulate Merlin^S13 phosphorylation to control Wnt signaling in meningioma cells.

Surgery is the mainstay of meningioma treatment and is often essential to relieve neurological symptoms from tumor mass effect[32]. Nevertheless, many meningiomas are diagnosed incidentally or with minimal presenting symptoms, and some incidentally-diagnosed meningiomas will not grow on long-term imaging surveillance[14].

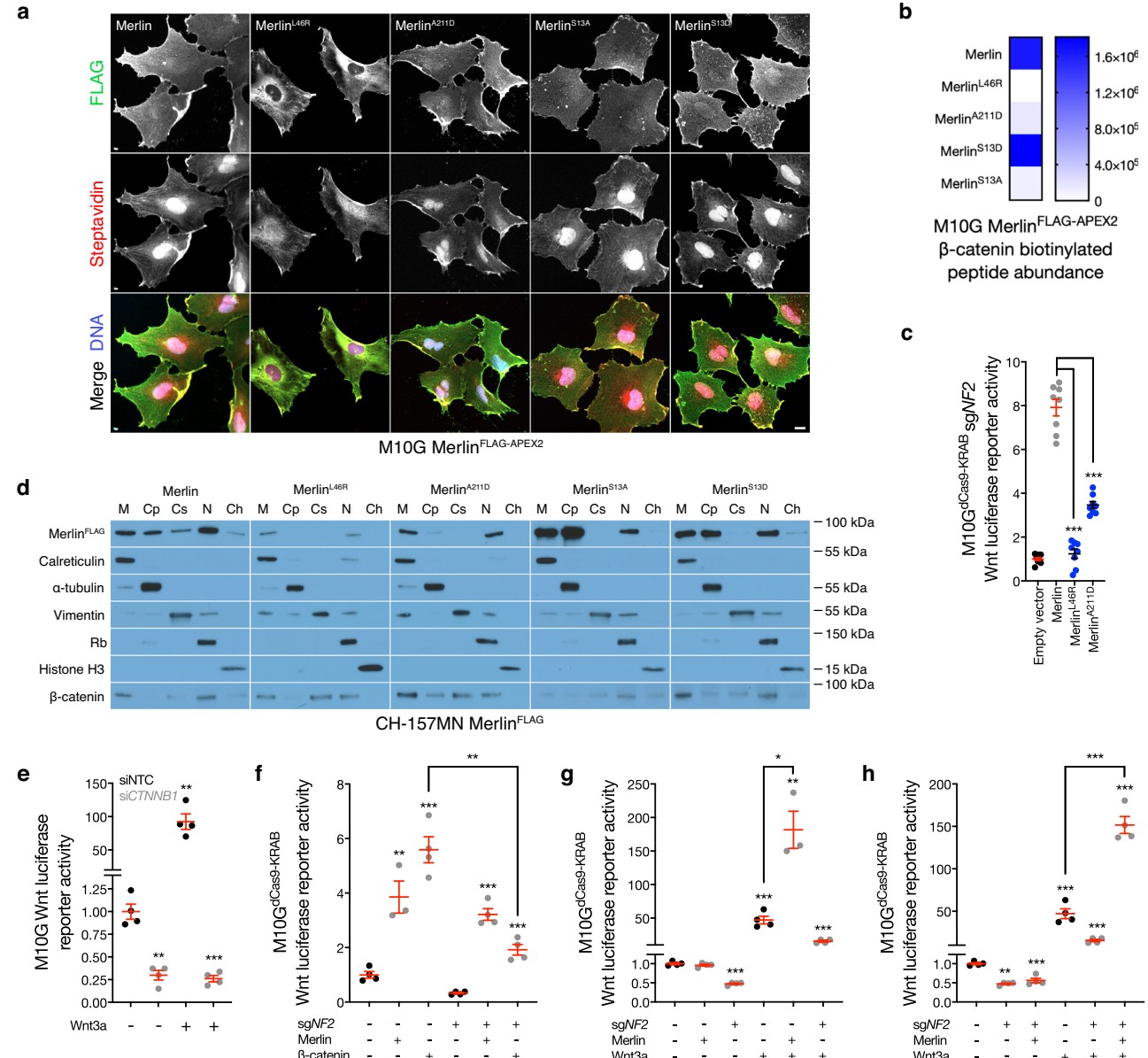

**Fig. 2 | Merlin regulates β-catenin localization in meningioma cells.**
**a** Immunofluorescence for FLAG (Merlin) or streptavidin in M10G meningioma cells expressing Merlin^FLAG-APEX2 constructs after proximity-labelling. Representative of five biological replicates. Scale bar, 10 μm. **b** Heatmap of β-catenin biotinylation peptide intensity from proximity-labeling proteomic mass spectrometry as in (**a**) (n = 3/condition). **c** TOP-Flash Tcf/Lef luciferase reporter assay in M10G^dCas9-KRAB meningioma cells expressing sgRNAs suppressing *NF2* (sg*NF2*) with or without rescue of Merlin^FLAG-APEX2 wildtype or cancer-associated missense constructs, n = 3. **d** Immunoblots for FLAG (Merlin) or β-catenin after biochemical fractionation of CH-157MN meningioma cells expressing Merlin^FLAG-APEX2 rescue constructs. Immunoblots for calreticulin, α-tubulin, vimentin, Rb, or histone H3 mark membrane (M), cytoplasmic (Cp), cytoskeletal (Cs), nuclear (N), or chromatin fractions (Ch), respectively. Representative of six biological replicates. **e** TOP-Flash Tcf/Lef luciferase reporter assay in M10G meningioma cells expressing non-targeted control

siRNAs (siNTC) or siRNAs suppressing β-catenin (si*CTNNB1*) with or without 24 h of Wnt3a treatment (100 ng/μl). Data are representative of four biological replicates. **f–h** TOP-Flash Tcf/Lef luciferase reporter assays in M10G^dCas9-KRAB meningioma cells expressing non-targeted control sgRNAs (sgNTC) or sg*NF2* with or without 24 h of Wnt3a treatment (100 ng/μl) in the presence or absence of β-catenin or Merlin overexpression or rescue. Data are representative of four biological replicates. **f** shows β-catenin overexpression fails to hyperactivate the Wnt pathway in the absence of Merlin. Data are representative of four biological replicates. **g**, **h** show Wnt stimulation is necessary for Merlin overexpression to hyperactivate the Wnt pathway. Data are representative of four biological replicates. Lines represent means, and error bars represent standard error of the means. **P ≤ 0.01, ***P ≤ 0.0001 (Student's t test, one sided). Source data are provided as a Source Data file.

These clinical observations underscore the unmet need for non-invasive, clinically-tractable biomarkers that predict meningioma outcomes and could be used to guide treatment de-escalation or imaging surveillance. Qualitative MRI features such as peritumoral edema, tumor calcification, tumor location, adjacent bone destruction, or irregular tumor margins can be associated with higher-grade meningiomas on preoperative imaging studies[33–35]. Although MRI can easily diagnose meningiomas, qualitative approaches are not reliable for distinguishing meningioma outcomes[36–39]. Quantitative apparent diffusion coefficient (ADC) hypointensity on diffusion-weighted MRI is prognostic for unfavorable meningioma outcomes[40], and may be associated with meningioma Wnt signaling[28], but biochemical mechanisms underlying meningioma imaging features are incompletely understood.

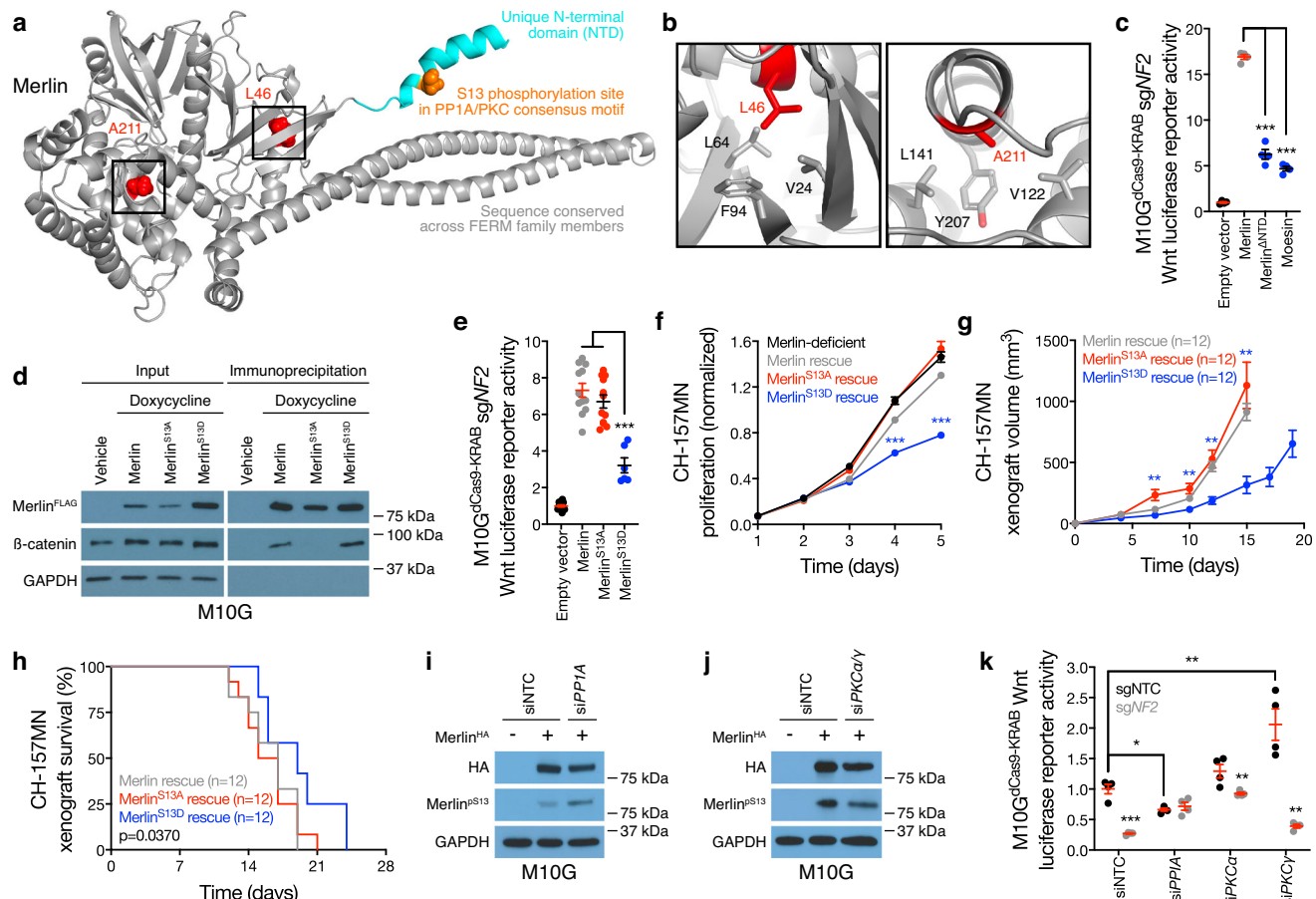

**Fig. 3 | Merlin dephosphorylation at Serine 13 activates meningioma Wnt signaling. a** Full-length structural model of Merlin, based on the crystal template 4RZJ. Red shows cancer-associated missense mutations. Cyan shows the NTD, which is not conserved in other FERM family members, containing a consensus PP1A/PKC phosphorylation motif and phosphorylation site (S13) in orange. **b** the 4RZJ X-ray structure of Merlin shows L46 and A211 in hydrophobic pockets that may be destabilized by charged cancer-associated missense mutations. **c** TOP-Flash Tcf/Lef luciferase reporter assay in M10G^(dCas9-KRAB) meningioma cells expressing sgRNAs suppressing *NF2* (sg*NF2*) with or without rescue of Merlin constructs or over-expression of the FERM family member Moesin. Data are representative of four biological replicates. **d** Immunoblots for FLAG (Merlin), β-catenin, or GAPDH before versus after FLAG immunoprecipitation from M10G meningioma cells with or without doxycycline-inducible Merlin^FLAG overexpression (1 μg/ml). **e** TOP-Flash Tcf/Lef luciferase reporter assay in M10G^(dCas9-KRAB) meningioma cells expressing sg*NF2* with or without rescue of Merlin wildtype or S13 unphosphorylatable (S13A) or phospho-mimetic (S13D) constructs. Representative of 3 experimental repeats. **f** CH-157MN meningioma cell MTT assays for cell proliferation with or without doxycycline-inducible Merlin rescue (100 ng/ml). Representative of four experimental repeats. **g** CH-157MN xenograft measurements in NU/NU mice with or without doxycycline-inducible Merlin rescue (20 μg/ml) as in (**f**). **h** Kaplan-Meier survival curve for CH-157MN xenograft overall survival in NU/NU mice as in **g** (log-rank test). **i, j** Immunoblots for HA (Merlin) or Merlin with phosphorylated S13 (Merlin^pS13) after Merlin immunoprecipitation from M10G cells with or without Merlin overexpression and concurrent expression of non-targeted control siRNAs (siNTC) or siRNAs suppressing PP1A or PKC isoforms. GAPDH immunoblots are shown from immunoprecipitation inputs as a loading control. Representative of four experimental repeats. **k** TOP-Flash Tcf/Lef luciferase reporter assay in M10G^(dCas9-KRAB) meningioma cells expressing non-targeted control sgRNAs (sgNTC) or sg*NF2* with concurrent siNTC or siRNA suppression of PP1A or PKC isoforms. Data are representative of four biological replicates. Lines represent means, and error bars represent standard error of the means. *$P \leq 0.05$, **$P \leq 0.01$, ***$P \leq 0.0001$ (Student's *t* test, one sided). Source data are provided as a Source Data file.

To study associations between meningioma ADC and tumor biology, 100 preoperative MRIs from meningiomas with available DNA methylation profiling and targeted exome sequencing of the *NF2* locus were retrospectively reviewed and imaging features were analyzed in the context of clinical follow-up data[7,13]. Meningioma ADC was dichotomized at the mean (Fig. 4a), showing ADC high meningiomas had favorable clinical outcomes compared to ADC low meningiomas (5-year local freedom from recurrence 90.3% versus 48.7%, *P* < 0.0001, log-rank test) (Fig. 4b). DNA methylation profiling controlled for artifacts from copy number variants (CNVs) reveals meningiomas are comprised of Merlin-intact, Immune-enriched, and Hypermitotic DNA methylation groups[7], which are concordant with groups and sub-groups of meningiomas derived from RNA sequencing or DNA methylation profiling integrated with RNA sequencing, CNVs, and SSVs[8,11,12]. Analysis of DNA methylation groups across ADC high versus ADC low meningiomas showed the majority of ADC high meningiomas were Merlin-intact (56% versus 16%, *P* < 0.0001, chi-squared test) (Fig. 4c). To determine if Wnt signaling was associated with meningioma ADC, β-catenin was suppressed in meningioma cells using shRNAs (Supplementary Fig. 7a), which inhibited Wnt signaling (Supplementary Fig. 7b), attenuated cell proliferation and tumor growth (Fig. 4d and Supplementary Fig. 7c), and prolonged overall survival compared to mice with meningioma xenografts expressing shNTC (Fig. 4e). MRI of Merlin-deficient meningioma xenografts showed Merlin^WT or Merlin^S13A rescue did not alter ADC, but ADC was increased with Merlin^S13D rescue or Merlin^WT rescue with concurrent suppression of β-catenin (Fig. 4f). Thus, meningioma ADC is inversely associated with Wnt pathway activation, and ADC high meningiomas have favorable clinical outcomes in human patients and preclinical xenograft models.

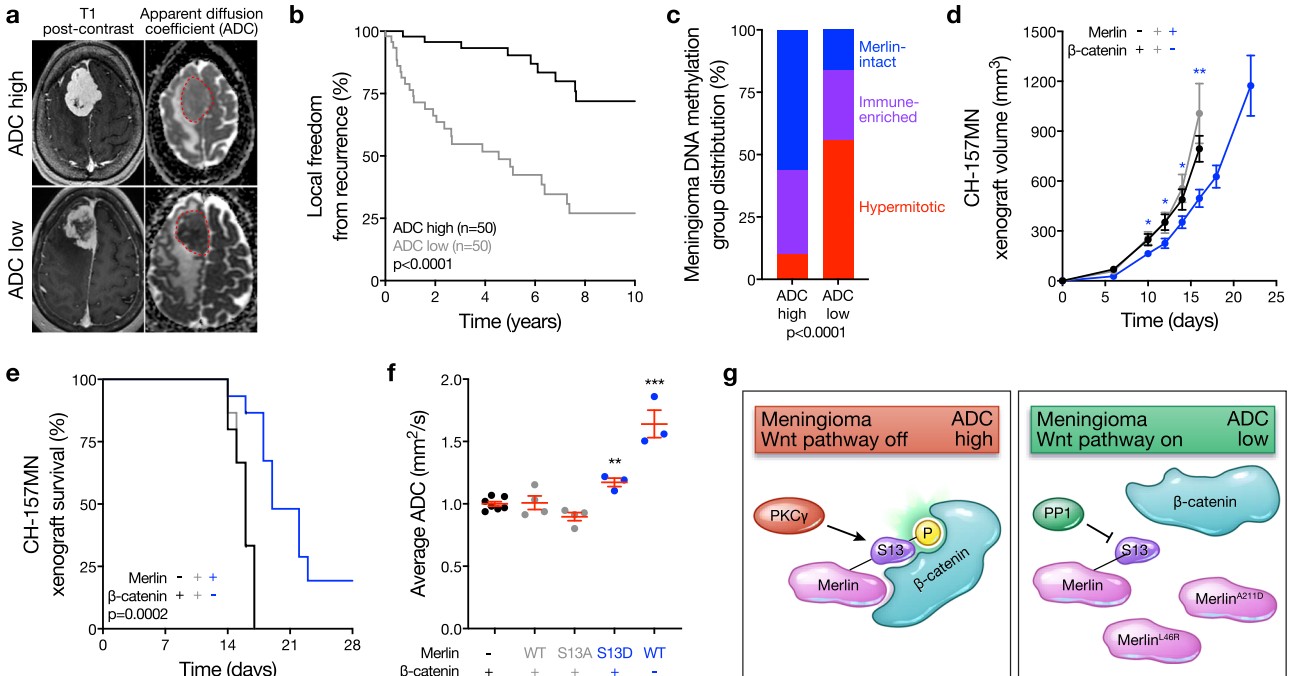

**Fig. 4 | High MRI apparent diffusion coefficient distinguishes Merlin-intact meningiomas with favorable clinical outcomes. a** Example brain magnetic resonance imaging (MRI) T1 post-contrast and apparent diffusion coefficient (ADC) maps. Red dotted lines show meningiomas on ADC maps. Representative of $n = 100$ meningiomas. **b** Kaplan-Meier survival curve for local freedom from recurrence in 100 human meningioma patients with pre-operative MRI analysis and post-operative DNA methylation profiling dichotomized at the mean normalized ADC (1.21, log-rank test). **c** Meningioma DNA methylation groups across ADC high versus ADC low strata from the 100 patients in (**b**) (Chi-squared test). **d** CH-157MN

xenograft measurements in NU/NU mice with or without doxycycline-inducible Merlin rescue or shRNA suppression of β-catenin ($n = 15$/condition). **e** Kaplan-Meier survival curve for CH-157MN xenograft overall survival in NU/NU mice as in (**d**) ($n = 15$/condition, log-rank test). **f** Normalized ADC from MRI of CH-157MN xenografts as in (**d**, **e**). **g** Model of meningioma Wnt signaling in the context of Merlin post-translational modifications, PP1A and PKC activity, cancer-associated missense mutants, and meningioma ADC. Lines represent means, and error bars represent standard error of the means. *$P \leq 0.05$, **$P \leq 0.01$, ***$P \leq 0.0001$ (Student's $t$ test, one sided). Source data are provided as a Source Data file.

## Discussion

Here we show Merlin drives meningioma Wnt signaling and tumor growth through a feed-forward mechanism that requires Merlin$^{S13}$ dephosphorylation to attenuate inhibitory interactions with β-catenin and activate the Wnt pathway (Fig. 4g). Integrating data from meningioma xenografts and patients, our results establish signaling pathways that underlie meningioma ADC as a potential imaging biomarker of Wnt signaling in Merlin-intact meningiomas with S13 phosphorylation and favorable clinical outcomes (Fig. 4g). These data shed light on how meningiomas can grow despite expression of Merlin, a canonical tumor suppressor, and provide a mechanistic basis for meningioma ADC, which has been previously proposed as a non-invasive biomarker of meningioma outcomes in humans[41]. Merlin-intact meningiomas tend to be benign and have the most favorable outcomes across molecular groups of human meningiomas[7,8,11,12]. Many meningiomas can be safely observed with serial imaging surveillance[14], but differentiating benign from aggressive meningiomas without subjecting patients to the morbidities associated with potentially unnecessary surgical treatments has been a barrier to improving clinical paradigms for patients with the most common primary intracranial tumor. To address this unmet need for patients, our data elucidate mechanisms underlying a clinically-tractable imaging biomarker that sheds light on meningioma biology and could be used to guide treatment de-escalation or imaging surveillance.

Our finding that Merlin post-translational modification can promote Wnt signaling is unexpected considering the well-described tumor suppressor functions of *NF2*. CNVs deleting *NF2* on chromosome 22q are early events underlying Immune-enriched or Hypermitotic meningioma tumorigenesis[7,28], but Merlin-intact meningiomas encode SSVs targeting *TRAF7*, *AKT1*, or *KLF4* that may be tumor-

initiating[17,18]. Our data show Merlin$^{S13}$ phosphorylation status and Wnt signaling modify Merlin-intact meningioma growth, but the way(s) in which this mechanism interacts with (or is influenced by) the myriad missense mutations in *TRAF7*, *AKT1*, *KLF4*, or other potential driver mutations that are enriched in Merlin-intact meningiomas remains to be established. Previous investigations have identified Wnt pathway activation across meningiomas with SSVs targeting *TRAF7*, *KLF4*, *PIK3CA*, *POLR2A*, the Hedgehog pathway, and even *NF2* and *SMARCB1*[25]. In support of the robustness of Wnt signaling across Merlin-intact meningiomas, re-analysis of RNA sequencing data from meningiomas with SSVs targeting *TRAF7* ($n = 8$), *PIK3CA* ($n = 5$), AKT1 ($n = 7$), or SMO ($n = 14$) revealed no difference in expression of Wnt target genes across these genetic contexts[13] (Supplementary Data 4). In meningiomas with adverse clinical outcomes and loss of *NF2*, the Wnt pathway may be activated by epigenetic silencing of endogenous Wnt inhibitors[24], but there does not appear to be an epigenetic basis of Wnt pathway activation in meningiomas with favorable clinical outcomes and expression of *NF2*[7]. Proximity-labeling proteomic mass spectrometry coupled with mechanistic and functional approaches demonstrate PKC and PP1A are important for this signaling mechanism in Merlin-intact meningiomas, but (1) other kinases or phosphatases, (2) other Merlin domains, or (3) other Wnt pathway members, including other mediators of β-catenin subcellular localization, such as CK1, GSK3β, JNK, β-TRcP, KDM2A, and TWA1[42], may also contribute to Wnt signaling in Merlin-intact meningiomas. In support of this hypothesis, we show (1) suppression of PP1A or PKC isoforms partially regulates Merlin$^{S13}$ phosphorylation (Fig. 3i, j), (2) epistatic Merlin$^{S13D}$ or Merlin$^{\Delta NTD}$ rescue partially restores Wnt signaling in meningioma cells (Fig. 3c, e), and (3) the non-canonical Wnt pathway regulators AMOT, AMOTL1, AMOTL2, DSG2, and DLG1[43–45] can be found in proximity to

Merlin alongside β-catenin in meningioma cells (Supplementary Data 2).

Crosstalk between Merlin and the Wnt pathway is complex, and interactions between Merlin and LRP6 can inhibit the Wnt pathway in non-cancer cell lines, schwannoma cells, tissues from patients with neurofibromatosis type 2, and xenopus embryos[46]. Loss of Merlin is associated with β-catenin phosphorylation and Wnt signaling in schwannoma cells[47], and in meningiomas the Wnt pathway can be activated by multiple mechanisms in tumors with unfavorable clinical outcomes[24]. Thus, our results may shed light on potential targets for future molecular therapies that could be used to treat molecularly-defined groups of meningiomas or schwannomas that are resistant to standard interventions. With respect to the putative feed-forward circuit we report, in unstimulated meningioma cells, we found (1) that knockdown of endogenous *NF2* negatively regulated Tcf/Lef transcription activity (Fig. 2g, columns 1 vs 3, and Fig. 2h, columns 1 vs 2), (2) that over-expression of exogenous *NF2* in cells lacking endogenous *NF2* could not rescue Tcf/Lef transcriptional activity (Fig. 2h, columns 2 vs 3) and (3) that over-expression of exogenous *NF2* in cells expressing endogenous *NF2* had no impact on Tcf/Lef transcriptional activity (Fig. 2g, columns 1 vs 2). However, after meningioma cell stimulation with Wnt3a, (4) knockdown of endogenous *NF2* resulted in a 50% loss of Tcf/Lef transcriptional activity (Fig. 2h, columns 4 vs 5), but (5) over-expression of exogenous *NF2* either with endogenous Merlin (Fig. 2g, columns 4 vs 5) or without endogenous Merlin (Fig. 2h, columns 5 vs 6) increased Tcf/Lef transcriptional activity above endogenous levels. Together, these data show that Merlin supports Wnt signaling at basal levels, but when endogenous Merlin is lost, its role in Wnt signaling cannot be rescued until the Wnt pathway is stimulated. These data further demonstrate that the Wnt pathway needs to be on for Merlin to promote Wnt signaling, and in the context of the other data in our study, our results suggest that Merlin is dephosphorylated as a result of Wnt pathway activation, which in turn promotes Wnt signal transduction through β-catenin and Tcf/Lef.

The Merlin NTD is unique compared to other FERM family members but is evolutionarily conserved across Merlin orthologs in higher eukaryotes. Previous studies show Merlin[S13] phosphorylation by MAP4K mediates angiogenesis[48], and Merlin serine 518 (S518) phosphorylation controls protein conformation[49–52], but the signaling pathways that are influenced by Merlin[S13] or Merlin[S518] phosphorylation are incompletely understood. More broadly (and in contrast to other FERM family members), it is unknown how tertiary conformational changes affect Merlin functions. Like other FERM family members, the open/closed conformation of Merlin requires interactions between the CTD and the FERM domain[51,53,54], which do not appear to be influenced by S13 phosphorylation on the flexible NTD (Fig. 3a).

This study should be interpreted in the context of its limitations, which include the paucity of preclinical models for studying meningioma biology. CH-157MN cells are a notoriously aggressive meningioma cell line that have been in culture for many years[22], and it is possible that the lack of phenotypic differences between CH-157MN xenografts with versus without Merlin expression was due to the malignant nature of these cells (Fig. 1b, c). To address this limitation, we validated our results using IOMM-Lee meningioma xenografts (Supplementary Fig. 2d, e), which recapitulate the histology of human meningiomas (Supplementary Fig. 2b) better than CH-157MN xenografts (Supplementary Fig. 1a). Moreover, we validated our finding that Merlin regulates the Wnt pathway in CH-157MN meningioma xenografts (Fig. 1d–f) using M10G meningioma cells (Figs. 1g, 2c, 2e–h. 3c–e, 3i–k), which were recently derived from a non-malignant meningioma[28]. We also tested the generalizability of these results using benign HEI-193 human schwannoma cells[30] (Supplementary Fig. 1b, c). More broadly, robust multi-institutional data demonstrate that human Merlin-intact meningiomas have better clinical outcomes than meningiomas with bi-allelic inactivation of *NF2*[7,8,11,12]. Thus,

despite the intrinsic challenges to modeling benign meningiomas in preclinical systems, the data presented here may provide a useful platform for future studies in humans. With respect to translation to human patients, intratumor heterogeneity can influence meningioma ADC measurements and correlates with intratumor Wnt signaling[28]. It is unknown if intratumor heterogeneity will be a barrier to adoption of non-invasive imaging biomarkers of meningioma biology, but emerging evidence suggests that intratumor heterogeneity is an important determinant of meningioma susceptibility to molecular therapy[55]. It is likely that multiple molecular mechanisms contribute to meningioma ADC, and future clinical trials that incorporate imaging and biomarker studies will be necessary to answer these questions. In the interim, our results identify a signaling pathway that underlies a potential imaging biomarker that could be used to guide treatment de-escalation or imaging surveillance for patients with favorable meningiomas.

## Methods
This study complied with all relevant ethical regulations and was approved by the UCSF Institutional Review Board (IRB #10-01141 and #18-24633). As part of routine clinical practice at both institutions, all patients who were included in this study signed a written waiver of informed consent to contribute data and tissue to research.

### Cloning
Plasmids encoding genes of interest were purchased from Addgene, or when unavailable, genes of interest were PCR amplified from cDNA. PCR products were cut using sticky-end restriction enzymes and ligated into plasmids with T4 ligase (NEB, Cat# M0202L). Ligated plasmids were transformed into Top10 or Stable II *E.coli*, colonies were isolated and expanded, and plasmid DNAs were sent for Sanger sequencing to confirm genes of interest. sgRNA sequences included *NF2* (5′-GGACTCCGCGCGCCTCTCAG-3′) and non-targeted control (5′-GTGCACCCGGCTAGGACCGG-3′). Pooled siRNA sequences included *PRKCA* (5′-UGGUUUACAUGUCGACUAA, UUAUAGGGAUCUGAAGUUA, GAAGGGUUCUCGUAUGUCA, UCACUGCUCUAUGGACUUA-3′), *PRKCG* (5′-GCCCGUAACCUAAUUCCUA, GGAGGGCGAGUAUUACAAU, GGGAGCGGCUGGAACGAUU, CAGAAGACCCGAACGGUGA-3′), *PP1CA* (5′-GCAAGAGACGCUACAACAU, GAGCAGAUUCGGCGGAUCA, CAUCUAUGGUUUCUACGA, GAACGACCGUGGCGUCUCU-3′), *CTNNB1* (Cell signaling, Cat# 6225S), and non-targeted control (5′-UGGUUUACAUGUCGACUAA-3′). shRNA sequences included *PRKCA* (5′-ATCAGCTCCGAAACTCCAAAG-3′), *PRKCB* (5′-TTCTCCCAATCAATATACCGG-3′), *PRKCG* (5′-TAAGGTCAGGGAGAATATCGG-3′), and *NF2* (5′-ATGTCAGTATCTTTGAAGTCG-3′).

### Cell culture
IOMM-Lee, CH-157MN, HEI-193, and HEK293T cells were purchased from ATCC and cultured in Dulbeco's Modified Eagle's Medium (DMEM) (Gibco, Cat# 11960069) supplemented with 10% Fetal Bovine Serum (FBS) (Gibco, Cat# 26140-079) and GlutaMAX (Gibco, Cat# 35050079). M10G cells were isolated from a primary human meningioma and cultured in 1:1 growth media comprised of DMEM F12 (Fisher Scientific, Cat# 11-320-082) and Neurobasal media (Gibco, Cat# 10888022) supplemented with 5% FBS, 1X GlutaMAX, 1 mM NEAA (Gibco, Cat# 11140050), N2 supplement (Gemini Bio-Products, Cat# 400-163), B27 (Gibco, Cat# 12587-010), 20 ng/ml epidermal growth factor (PeproTech, Cat# AF-100-15), and 20 ng/ml fibroblast growth factor (PeproTech, Cat# AF-100-18B). Cells were split 1:10–1:5 every 3–4 days, or when confluency reached 95%. Routine testing for contamination of cell cultures was performed using the Cell Culture Contamination Detection Kit (Molecular Probes, Cat# C-70208) and LookOut Mycoplasma PCR Detection Kit (Sigma, Cat# MP0040). Cell lines were authenticated using STR profiled once per year, which includes PCR amplification of 9 STR loci plus Amelogenin using the Promega GenePrint 10 System, Fragment Analysis with ABI 3730XL

DNA Analyzer, comprehensive data analysis with ABI Genemapper software, and final verification using supplier databases including ATCC and DSMZ.

## Lentiviral transduction

For virus production, HEK293T cells were seeded into 10 cm plates at $2 \times 10^5$ cells/ml in 10 ml. Lentiviral packaging plasmids pMD2.G and psPAX2 were transfected using the Mirus Trans-IT (Mirus Bio, Cat# MIR 2225) virus transfection reagent. For transduction, HEK293T viral media was filtered through a 0.45 μm PET syringe filter and incubated with host cells with 1:1 host cell media and 10 nM polybrene (EMD Millipore, Cat# TR-1003-G). For antibiotic selection, cells were treated with blastocydin (Goldbio, Cat# B-800-25) or puromycin (Invivogen, Cat# ant-pr-1) 48 h after transduction. Cells expressing dCas9-KRAB-BFP were washed 48−96 h after transduction, trypsinised and pipette into a single cell suspension before filtering through a 40 μM cell strainer (Falcon, Cat# 352340) and sorted for all BFP+ cells using FACS (Sony SH800). Approximately $5 \times 10^5$ cells were sorted and reseeded back into a 10 cm plate and expanded to T75 flasks. Once this culture approached confluency, cells were sorted again for the top 10% of BFP + cells and left to expand. To introduce sgRNAs into cells expressing dCas9-KRAB-BFP, plasmids containing sgRNAs were transfected with pMD2.G and psPAX2, and sgRNA-positive cells were selected using puromycin as above.

## Mouse xenografts

Xenograft experiments were performed by injecting 3 million human meningiomas cells, either CH157-MN or IOMM-Lee, subcutaneously into the flank of 4−6-week-old NU/NU female mice (Envidigo) using an 25 g needle (BD, Cat# 14-826-49). Meningiomas arise in patients of both sexes and the phenotypes reported here do not have differences between sexes in humans, so a decision was made to use female mice to minimize altercations between animals subjects. Tumors were usually visible 7−10 days after injection, and once tumors became measurable, volume was calculated using (1.5 × width) × length. For shRNA induction, cells were pre-treated with 3 mM IPTG (Sigma Aldrich, Cat# I6758) 10 days prior to injection. To induce plasmid or shRNA expression, mice were treated with or without 200 μg/ml doxycycline (Sigma-Aldrich, Cat# D9891), with or without 10 M IPTG (Research Products International, Cat# I56000), in cage water that was changed every 2−3 days. Kaplan-Meyer curves were created by recording deaths at protocol-defined endpoints of 50% ulcerated tumor or tumor >2000 mm³. Tumors were processed for single-cell dissociation and single-cell RNA sequencing, QPCR, or immunoblotting.

All animal care and experimental procedures were in accordance with federal policies and guidelines governing the use of animals and were approved by the University of California San Francisco's (UCSF) Institutional Animal Care and Use Committee (IACUC). The IACUC is in full compliance with the 8th edition of The Guide for the Care and Use of Laboratory Animals. UCSF has an AAALAC-accredited animal care and use program. Mice were housed in solid-bottomed cages containing autoclaved paper chips in individually ventilated cages. Animals had continuous access to irradiated food and water purified by reverse osmosis and UV lighting. The housing room was maintained at 68−74 °F with 30−70% relative humidity. All cages were maintained in an SPF barrier facility from which dirty bedding sentinel mice were tested quarterly. All sentinels were found to be seronegative for mouse hepatitis virus, pneumonia virus of mice, mouse parvovirus, minute virus of mice, epizootic diarrhea of infant mice, Theiler's murine encephalo-myelitis virus, ectromelia and were free of ectoparasites and endoparasites. Mice were observed daily by animal care staff for any clinical abnormalities and were euthanized in accordance with American Veterinary Medical Association guidelines using a 2-step procedure involving inhaled carbon dioxide followed by cervical dislocation.

## Xenograft lysis

Tumors were dissected into small chunks and allocated for either single-cell RNA sequencing, RNA extraction for cDNA synthesis and QPCR, or protein extraction and immunoblotting. For RNA extraction for cDNA synthesis and QPCR, or protein extraction and immuno-blotting, tumor chunks were placed in a 2 ml Eppendorf tube with a 7 mm metal bead and 350 μl of RLT lysis buffer for RNA extraction, or 300 μl RIPA buffer for protein. Tubes were shaken in a TissueLyser for 2 min at 30 Mhz, lysate was cleared by spinning in a benchtop centrifuge at 16,000 ×g for 5 min at 4 °C. Supernatant was removed and used for downstream analyses.

## Immunoblotting

Cells were lysed in either RIPA (150 mM sodium chloride, 50 mM Tris-HCl [pH8], 1% NP40, 0.25% sodium deoxycholate, 0.1% SDS) or JIES buffer (100 mM NaCl₂, 20 mM Tris-HCl [pH7.4], 5 mM MgCl₂, 0.5% NP40) containing protease inhibitor (Complete-Mini, Sigma Aldrich, Cat# 11836170001) and phosphatase inhibitor (Phos-STOP, Roche, Cat# 04906837001) and quantified using the Bradford assay (Bio-Rad, Cat# 5000205). Normalized protein lysates were run on 4−20% Mini-PROTEAN TGX gels (Bio-Rad, Cat# 4561096) at 250 V for 30 min, and transferred onto nitrocellulose membrane (Bio-Rad, Cat# 1620094) at 100 V for 30 min at 4 °C sandwiched between two sheets of filter paper. Membranes were blocked in 5% BSA in TBST for 1 h at room temperature, probed with primary antibodies at the indicated concentrations for either 1 h at room temperature or overnight at 4 °C, and incubated with HRP-conjugated secondary antibodies (1:2000 dilution) for 1 h at room temperature. Primary and secondary antibodies were FLAG (Sigma-Aldrich, Cat# F7425, 1:500−2000), DYKDDDDK (Cell Signaling, Cat# 14793S, 1:2000), HA (Cell Signaling, Cat# 2999S, 1:2000), GAPDH (Thermo Fisher Scientific, Cat# MA515738, 1:5000), Merlin (Abcam, Cat# ab88957, 1:1000), α-Tubulin (Sigma-Aldrich, Cat# T5168, 1:5000), calreticulin (Abcam, Cat# ab92516, 1:10,000), vimentin (Abcam, Cat# ab8069, 1:10,000), Rb (Cell Signaling, Cat# 9309S, 1:5000), Histone H3 (Thermo, Cat# 702023, 1:5000), GST (Cell Signaling, Cat# 2622S, 1:1000), and β-catenin (BD Biosciences, Cat# 610153, 1:1000). Anti-phospho-Serine13 (1:250−500) was a custom antibody developed by Thermo Fisher Scientific using rabbits that were immunized with a synthetic Merlin phospho-peptide sequence (CSRMSFS(pS)LKRKQP-amide). Chemiluminescence was detected with Pierce ECL (Thermo Scientific, Cat# 32209) and developed on autoradiography film after incubation with rabbit (Cell signaling, Cat# 7074, 1:2000) or mouse (Cell signaling, Cat# 7076, 1:2000) HRP-conjugated secondary antibodies. For phospho-peptide competition assays, four identical immunoblot membranes were incubated with either (1) HA-HRP, (2) 4 μg phS13 Merlin Ab, (3) 4 μg phS13 Merlin Ab with 20 μg phosphorylated peptide, or (4) 4 μg phS13 Merlin Ab with 20 μg unphosphorylated peptide. All membranes were incubated with secondary antibody and developed at the same time.

## Immunohistochemistry and light microscopy

Deparaffinization and rehydration of 5 μm FFPE meningioma xenograft sections and hematoxylin and eosin staining were performed using standard procedures. Immunostaining was performed on 5 μm FFPE meningioma xenograft sections using an automated Ventana Discovery Ultra staining system. Immunohistochemistry was performed using rabbit monoclonal Ki-67 (Ventana, clone 30-9, Cat# 790-4286, 1:6) with incubation for 16 min. Histologic and immunohistochemical features were evaluated using light microscopy on an BX43 microscope with standard objectives (Olympus). Images were obtained and analyzed using the Olympus cellSens Standard Imaging Software package.

## Single-cell dissociation

After mice were euthanized using CO₂, tumors were freshly harvested and sliced into small pieces using two #10 scalpels on a glass plate.

Initial enzymatic dissociation was performed using 0.1 mg/ml Colagenase Type 7 (Worthington Biochemicals, Cat# LS005332) at 37 °C for 30 min with rotation, followed by second enzymatic digestion with 0.25% trypsin (Thermo Fisher Scientific, Cat# 25200114) at 37 °C for 10 min. Red blood cells were removed in 1X RBC lysis buffer (Invitrogen, Cat# 00-4300-54) at room temperature for 10 min. To generate single-cell suspensions, cells were sequentially filtered through 70 μm and 40 μm filters. Finally, 8000 cells were calculated by mixing cell suspensions 1:1 with tryphan blue and counting with a Life Technologies Countess II.

### Single-cell RNA sequencing and analysis

Cells were diluted to 1000 cells per microliter as per the 10X Genomics protocol for optimal single-cell processing. Single cells were passed through a 10X Genomics Chromium Controller and libraries were created using the Chromium Single Cell 3′ Library & Gel Bead Kit v3 (10X Genomics, Cat# 1000121) including primer annealing, cDNA amplification, fragmentation, size selection, and index annealing according to the manufacturers' protocol. After successful completion of steps 2.4 and 3.6, libraries were diluted 1:10 and quantified on a D5000 (Agilent, Cat# 5067-5592 and Cat# 5067-5593) or D1000 High Sensitivity TapeStation (Agilent, Cat# 5067-5584 and Cat# 5067-5585), respectively. Libraries were pooled and sequenced on a NovaSeq S4 6000 at the UCSF Center for Advanced Technology.

Demultiplexing, identification of empty droplets, removal of duplicates, and alignment to the human or mouse reference genomes was performed using CellRanger. Count matrices were selected using cells with more than 50 unique genes, less than 8000 unique genes, and with less than 20% of genes assigned as mitochondrial. Data were processed using Seurat in RStudio using SCTransform. Dimensionality reduction was performed using principal component analysis. UMAP was performed on the reduced dimensionality data using the minimum distance of 0.2 and a Louvain clustering resolution of 0.8. Differentially expressed genes were identified with Wilcoxon Rank Sum test in Seurat. Cell states were identified using top differentially expressed genes.

### Luciferase assay

Cells were seeded into 24-well plates at $0.5 \times 10^5$ cells/ml in 500 μl. The following day they were transfected with pRL-TK TOP-Flash Tcf/Lef luciferase reporter with or without additional genes of interest, as indicated, for 48 h. 24 h before experimentation, cells were treated with or without 200 ng/ml Wnt3a (R&D Systems, Cat# 5036-WN). To detect luciferase activity, cells were lysed in passive lysis buffer using the Dual Luciferase Kit (Promega, Cat# G4100), lysed for 15 min with rotation at room temperature, and luciferase activity was detected with GLO-Max Promega plate reader.

### Quantitative polymerase chain reaction

QPCR primers were designed using PrimerBank (https://pga.mgh.harvard.edu/primerbank/). RNA was isolated using the QiaGen RNeasy prep kit (Cat# 74106). cDNA was prepared using 1000 ng of RNA and iScript reverse transcriptase (Bio-Rad, Cat# 1708891). QPCR was performed using PowerUp SYBR Green (Thermo Scientific, Cat# A25742) in a Life Technologies QuantStudio 6 Flex Real-Time PCR system. Relative gene expression was calculated using the ΔΔCt method against a control gene, *GAPDH*. QPCR primers used were NF2_F (5′-TT GCGAGATGAAGTGGAAAGG-3′), NF2-R (5′-CAAGAAGTGAAAGGTGA CTGGTT-3′), PP1A_F (5′-ACTACGACCTTCTGCGACTAT-3′), PP1A_R (5′-A GTTCTCGGGGTACTTGATCTT-3′), PRKCA_F (5′-GTCCACAAGAGGT GCCATGAA-3′), PRKCA_R (5′-AAGGTGGGGCTTCCGTAAGT-3′), PRKCG_F (5′-AGCCACAAGTTCACCGCTC-3′), PRKCG_R (5′-GGACA CTCGAAGGTCACAAAT-3′), CTNNB1_F (5′-CATCTACACAGTTTGATG CTGCT-3′), CTNNB1_R (5′-GCAGTTTTGTCAGTTCAGGGA-3′), AXIN2_F (5′-TACACTCCTTATTGGGCGATCA-3′), AXIN2_R (5′-TTGGCTACTCGT

AAAGTTTTGGT-3′), DKK_F (5′-ATAGCACCTTGGATGGGTATTCC-3′), DKK_R (5′- CTGATGACCGGAGACAAACAG-3′), GAPDH_F (5′-GTCT CCTCTGACTTCAACAGCG-3′), and GAPDH_R (5′-ACCACCCTGTTGCT GTAGCCAA-3′).

### Proximity-labeling proteomic mass spectrometry

M10G cells stably express pLV.APEX2 constructs encoding wildtype or variant Merlin constructs were seeded onto 5 x 15 cm plates. Cells were treated with 0.1 μg/ml doxycycline to induce Merlin expression 24 h before APEX labelling. For labelling, 0.5 mM biotin-phenol (Berry and Associates, Cat# BT1015) was added to each plate for 30 min at 37 °C, and 1 mM $H_2O_2$ (Sigma Aldrich, Cat# H1009) was added to cell media on ice for 30 s to initiate the reaction. Media were replaced with quenching media (10 mM Sodium Ascorbate, 1 mM Azide, 5 mM Trolox) for two washes. For mass spectrometry, cells were scraped and pelleted for biotin/streptavidin precipitation as previously described[7]. Cell pellets were resuspended in 8 M urea, 0.1 M ammonium bicarbonate (ABC), 150 mM NaCl, 1x mini-cOmplete protease inhibitors, 1x phosSTOP phosphatase inhibitor, 4 mM TCEP. Cells were then lysed by three rounds of 30 s probe sonication at 20% amplitude with a 10 s break. Lysates were then cleared by centrifugation, and proteins alkylated by the addition of 10 mM iodoacetamide for 30 min at RT in the dark. The akylation reaction was quenched by the addition of 10 mM DTT for 30 min at RT in the dark. Lysates were then diluted with 0.1 M of ABC pH 8 to a final urea concentration of 1.8 M. Proteolytic cleavage was then performed by the addition of Lys-C in a 1:50 enzyme to substrate and incubation at 37 °C overnight with agitation. The following day, 1 mg of digested lysate was desalted on a 50 mg C18 SepPak cartridge and the resulting eluant was dried by vacuum centrifugation. Next, the dried peptides were enriched for phosphorylated peptides by incubation with Ni-NTA Superflow beads that had been stripped with EDTA and loaded with iron chloride. The beads were washed four times with 80% ACN, 0.1% TFA to remove unphosphorylated peptides, followed by elution of the enriched phosphopeptides by incubation of the beads with 500 mM phosphate pH 7. The eluted phosphopeptides were then desalted on C18 pipette tips and resuspended in 0.1% formic acid prior to injection into the mass spectrometer.

For all MS experiments, mobile phase B was 0.1% FA in 80% ACN, and mobile phase A was 0.1% FA in $H_2O$. For proximity labeling experiments, gradient began at 5% B, before increasing to 25% B over 36 min. Mobile phase B then increased to 36% over 45 min before finishing with a 10 min wash at 90% B. The entire gradient required 90 min. Mass spectrometry scans were acquired in a data-dependent manner using a trybrid Orbitrap Fusion Lumos from Thermo Scientific. MS1 scans were taken at a resolution of 120,000 with a maximum injection time of 50 ms and a normalized AGC target of 250% over a scan range of 350–1350 m/z. Ions with a charge state of 2–5 were isolated using a 0.7 m/z window and fragmented using HCD with a normalized collision energy of 32% and dynamic exclusion within a 10 ppm range for 30 s to prevent repeat sequencing. Fragment (MS2) scans were collected at a resolution of 30,000 with a maximum injection time of 18 ms and a normalized AGC target of 300%. MS2 scans were collected from 200 to 1200 m/z with a 2 s cycle time between MS1 scans. Raw files were searched in MaxQuant using label-free quantitation with match between runs against the reviewed human proteome including isoforms, downloaded from Uniprot on October 22, 2020. Carbamidomethylation of cysteines was included as a fixed modification, with oxidation of methionine and N-terminal acetylation included as variable modifications. All other parameters were set to the default. For phosphoproteomic experiments, gradient began at 2% B, before increasing to 20% B over 85 min. Mobile phase B then increased to 32% over 20 min before finishing with an 8 min wash at 90% B and 1 min equilibration at 2% B. The entire gradient required 120 min. Mass spectrometry scans were acquired in a data-dependent

manner using Orbitrap Q-Exactive Plus from Thermo Scientific. MS1 scans were taken at a resolution of 70,000 with a maximum injection time of 100 ms and an AGC target of 3e6 over a scan range of 300–1500 m/z. Ions with a charge state >1 were isolated using a 2.2 m/z window and fragmented using HCD with a normalized collision energy of 27% and an automatic dynamic exclusion. Fragment (MS2) scans were collected at a resolution of 17,500 with a maximum injection time of 120 ms and an AGC target of 1e5. MS2 scans were collected from 200 to 2000 m/z with loop-controlled cycle time of 15 MS2 scans between consecutive MS1 scans. Raw files were searched in MaxQuant using label-free quantitation with match between runs and lysC as the digesting protease. The files were searched against the canonical-reviewed human proteome. Carbamidomethylation of cysteines was included as a fixed modification, with oxidation of methionine, n-terminal acetylation, and phosphorylation at STY included as variable modifications. All other parameters were set to the default.

## Immunofluorescence and confocal microscopy
Cells were seeded onto glass coverslips in 24-well plates. 24 h prior to fixing, cells were treated with 0.1µg/ml doxycycline and an APEX labeling reaction was performed as described above. Quenched cells were fixed in 4% PFA for 10 min at room temperature (Electron Microscopy Sciences, Cat# 15710) and stained using DYKDDDDK primary antibody (Cell Signaling, Cat# 14793S, 1:2000) and anti-mouse conjugated to Alexa Fluor 488 secondary antibody (Thermo Fisher Scientific, Cat# A21202, 1:2000), streptavidin conjugated to Alexa Fluor 647 (Thermo Fisher Scientific, Cat# S21374, 1:1000), and Hoechst (Invitrogen, Cat# H3570, 1:10,000). Coverslips were mounted onto slides using Prolong Diamond anti-fade mounting media (Thermo Fisher Scientific, Cat # P36965) and imaged on a Zeiss LSM800 fluorescence microscope.

## Plasmid overexpression
Cells were seeded into 6 cm plates at $0.5 \times 10^5$ cells/ml for 24 h before transfection. Transfection solution consisting of 500 µl Opti-MEM (Thermo Fisher Scientific, Cat# 51985091), 2 µg plasmid DNA, and 7.5 µl FuGene (Promega, Cat# E2311) was incubated at room temperature for 20 min before adding to cells. Media was changed the following day and cells were harvested 48 h after transfection for experimentation. Preparations were scaled appropriately according to the size of the plate.

## Biochemical fractionation
Cells with conditional expression of pLV.APEX2-Merlin constructs were seeded into 10 cm plates at $0.5–2 \times 10^5$ cells/ml. The following day, plates were treated with 0.1µg/ml doxycycline for gene induction. 48 h after induction, plates were washed with 4 ml ice-cold PBS and then scraped with 4 ml of PBS, transferred into a 15 ml falcon tube, and spun at $350 \times g$ for 5 min to pellet cells. Supernatant was discarded and cell pellets were resuspended in 1 ml ice-cold PBS and spun again at $350 \times g$ for 5 min. 20 µl of packed cell pellet from each sample was processed with the subcellular fractionation kit (Thermo Cat# 78840). After an initial lysis in 200 µl of cytoplasmic extraction buffer for 10 min rotating at 4 °C, soluble and insoluble fractions were separated by centrifugation for 5 min at $500 \times g$. Supernatant was transferred to a new tube and assigned as the cytoplasmic fraction and the cell pellet and tube were washed with 300 µl ice-cold PBS. PBS was discarded and the cell pellet was lysed again in 200 µl membrane extraction buffer with incubation at 4 °C for 10 min with rotation. Soluble and insoluble fractions were separated by centrifugation at $3000 \times g$ for 5 min at 4 °C. Supernatant was transferred to a new tube and assigned as the membrane fraction. The cell pellet and tube were washed with 300 µl ice-cold PBS to remove residual proteins. PBS was removed and nuclear fraction was isolated in nuclear extraction buffer for 30 min with rotation at 4 °C. Soluble and insoluble proteins were separated with centrifugation at $5000 \times g$ for 5 min at 4 °C. The supernatant was transferred to a new tube and

assigned as the nuclear fraction while the pellet and tube were rinsed with ice-cold PBS. PBS was carefully removed, and the chromatin fraction was extracted using 100 µl of chromatin extraction buffer (92 µl nuclear extraction buffer with 5 µl of 100 mM CaCl₂ and 3 µl micrococcal nuclease) for 15 min at room temperature. Soluble and insoluble fractions were separated with centrifugation at $16,000 \times g$ for 5 min. The soluble fraction was transferred to a new tube and assigned as the chromatin fraction. The cell pellet and tube were washed again with ice-cold PBS to remove contaminating proteins. Finally, the cytoskeletal fraction was extracted in pellet extraction buffer for 10 min at room temperature, centrifuged at $16,000 \times g$ for 5 min, and the supernatant was removed and assigned as the cytoskeletal fraction. All buffers were supplemented with protease inhibitor (Complete-Mini, Sigma Aldrich, Cat# 11836170001) and phosphatase inhibitor (Phos-STOP, Roche, Cat# 04906837001). After successful lysis of all subcellular fractions, the Bradford assay was used to calculate protein concentration and all samples were normalized before immunoblot. α-Tubulin (Sigma-Aldrich, Cat# T5168, 1:5000), calreticulin (Abcam, Cat# ab92516, 1:10,000), vimentin (Abcam, Cat# ab8069, 1:10,000), Rb (Cell Signaling, Cat# 9309S, 1:5000) and Histone H3 (Thermo, Cat# 702023, 1:5000) were used as controls for each subcellular compartment.

## siRNA knockdown
On day 1, reverse transfection of siRNA was achieved using RNAi-MAX (Invitrogen, Cat# 13778075). For each 6 cm plate, solution A (250 µl Opti-MEM, 25 nM siRNA) and solution B (250 µl Opti-MEM and 5 µl RNAiMAX) were made independently. The solutions were incubated at room temperature for 5 min and then combined, vortexed, and incubated for an additional 20 min before adding to cells at $0.2 \times 10^5$ cells/ml for seeding. Media were changed on day 2. RNAi-Max knockdown was repeated on day 3 followed by media change on day 4. For experiments which included plasmid overexpression, FuGene (Promega, Cat# E2311) transfection and simultaneous reseeding of cells were performed on day 5 for 48 h. Cells were harvested 7 days after the first siRNA transfection for downstream analysis.

## Structural modeling
To obtain a full-length 3D representation of the Merlin human protein, a structural model was generated using the Robetta server (https://robetta.bakerlab.org/) and the solved structure of human Merlin-FERM as a template (PDB 4ZRJ). The model was inspected using pymol to rationalize the role of disease-associated mutations. Robetta provides a fully automated modeling procedure exploiting both ab initio and comparative models of protein domains. Comparative models were built from structures detected and aligned by HHSEARCH, SPARKS, and Raptor. Loop regions were assembled from fragments and optimized to fit the aligned template structures.

## Immunoprecipitation
Cells were lysed in JIES buffer (100 mM NaCl₂, 20 mM Tris-HCl [pH7.4], 5 mM MgCl₂, 0.5% NP40) containing protease inhibitor (Complete-Mini, Sigma Aldrich, Cat# 11836170001) and phosphatase inhibitor (Phos-STOP, Roche, Cat# 04906837001) then left at 4 °C for 1 h with rotation. Protein concentration was normalized using the Bradford assay (Bio-Rad, Cat# 5000205) and incubated with pre-washed HA (Sigma-Aldrich, Cat# A2095, 30 µl per IP) or FLAG (Sigma-Aldrich, Cat# M8823, 30 µl per IP) antibody-bound beads. The sample/bead slurries were left to rotate at 4 °C for 4 h before washing four times in JIES buffer supplemented with protease and phosphatase inhibitors as described above. Bound proteins were eluted from beads by boiling in 30 µl 2x Laemmli buffer and separated by gel electrophoresis as described above.

## In vitro proliferation assay
Cells were seeded in quadruplicate at 2000 cells per well in $5 \times 96$-well plates in 100 µl of cell culture media. Each day after seeding a Cell-Titer

Glo MTT (Promega, Cat# G4100) assay was performed to observe cell proliferation day-by-day. Cells were treated with 15 µl of dye solution and left to incubate at 37 °C and 5% $CO_2$. After 1 h, reactions were quenched using Stop solution and plates were left in a 37 °C and 5% $CO_2$ incubator overnight. Readings were taken the following day at an absorbance of 560 nm on a Promega GloMAX Discover plate reader.

## Recombinant protein purification

Gene fragments were cloned into pGEX-4T1 (Cytivia, Cat# 28-9545-49) and validated using Sanger sequencing. For protein expression, constructs were heat shocked into BL21 bacteria (NEB, Cat# C2530H) for 10 s at 42 °C, then grown at 37 °C overnight with shaking in 5 ml of 2X LB with 50 ug/ml Ampicilin (Caymen Chemicals, Cat# 14417). After 15 h, 1 ml of overnight culture was diluted in 50 ml fresh 2x LB with Ampicillin. Once optical density reached 0.5–1, cultures were treated with 1 mM IPTG (Sigma, Cat# 16758) for an additional 3 h at 37 °C with shaking. Cell cultures were pelleted in 12 ml aliquots at max speed in a Heraeus X1R refrigerated benchtop centrifuge for 10 min (Thermo Scientific). To lyse bacterial cells, pellets were resuspended in 600 µl PBS, sonicated in a CO-Z Digital Pro+ bench top sonicator for 5 × 1 min with 30 s rests on ice. Next, 30 µl of 20% TritonX-100 (Sigma, Cat# X100) was added and left rotating for 30 min at 4 °C. For GST purification, lysed bacterial cell lysate was run through a GST SpinTrap Column (Cytivia, Cat# 28-9523-59) and eluted with 10 mM reduced glutathione (Thermo, Cat# A18014-14). Successful protein expression and purification were tested using polyacrylamide gel electrophoresis (Bio-Rad, Cat# 456-1096) and Coomassie stains (Abcam, Cat# ab119211) or immunoblot.

## In vitro binding assay

FLAG-tagged β-catenin was transfected into HEK293T cells. After 48 h, cells were washed, scraped and lysed in JIES buffer with 30 min rotation at 4 °C. FLAG-tagged β-catenin was immobilized on magnetic FLAG M2 beads (Sigma, Cat# M8823) rotating for 2 h at 4 °C. Beads were then washed 3× with JIES buffer containing protease and phosphatase inhibitors. After the final wash, beads were resuspended in 100 µl JIES buffer with equal amounts of GST-tagged proteins. From this 100 µl mixture, 25 µl was kept for input sample, and 375 µl buffer was added to the immunoprecipitation beads described in the previous section while rotating at 4 °C. After 1 h, beads were washed 3× with buffer and eluted in 30 µl 2× Laemmli buffer. For immunoblot analyses, 10 µl of each input and 3.3 µl of each immunoprecipitation sample was loaded.

## Bulk RNA sequencing and analysis

Library preparation was performed using the TruSeq RNA Library Prep Kit v2 (#RS-122- 2001, Illumina) and 50 bp single end reads were sequenced on an Illumina HiSeq 2500 or NovaSeq to a minimum depth of 25 M reads per sample at Medgenome, Inc. Sequencing reads were evaluated for quality metrics through FastQC, followed by an aggregate assessment using MultiQC. Adapter sequences and bases not meeting the quality threshold score (<30) were removed from both the 3' and 5' ends using Cutadapt. Sequences that resulted in a length shorter than 20 bases post-trimming were discarded. The remaining quality reads underwent alignment to the GRCh38 reference genome utilizing the HISAT2 v2.2.0 aligner, employing default alignment parameters. Alignment outputs were configured to generate files sorted by coordinate. The aligned reads were then quantified at the exon level to generate a gene expression matrix, using featureCounts v2.0.3 from the Subread package, with exon specified as the feature type and gene_id as the grouping attribute. The derived gene expression count matrix was further processed in R v4.3.1, using the DESeq2 package v1.36.0 for differential expression analysis. A filtering step was incorporated to exclude genes with counts lower than ten across all samples. Normalization and variance stabilization of count data were achieved through DESeq2's variance-stabilizing transformation (VST). Additional visualization and manipulation of the data were facilitated by the ggplot2 and pheatmap packages within R.

Gene Set Enrichment Analysis preranked was performed to identify pathways enriched in differentially expressed genes. The gene rank scores were calculated using the formula: SIGN(log2FC) × −log10($p$ value). Pathways were delineated by the gene set file Human_GOBP_AllPathways_no_GO_iea_April_02_2023_symbol.gmt, which is periodically updated and maintained by the Bader laboratory. Positive and negative enrichment profiles were achieved through 2000 permutations. The gene set size was constrained between 10 to 500 for the analysis. Pathway analysis results were visualized with EnrichmentMap through Cytoscape. Parameters set for nodes included an FDR $q$ value of less than 0.01, a $p$ value of less than 0.01, and nodes that share gene overlaps with a Jaccard + Overlap Combined (JOC) threshold of 0.375. Such nodes were interconnected with a blue line (edge) to formulate network maps. Clusters of analogous pathways were pinpointed and labeled utilizing AutoAnnotate in Cytoscape. This app incorporates a Markov Cluster algorithm, connecting pathways by mutual keywords in their description. The resulting groups of pathways are designated as the major pathways in a circle.

## Clinical magnetic resonance imaging

All patients underwent MRI examinations on a 3T Discovery MR750 scanner (GE Healthcare) using an eight-channel phased-array head coil prior to surgical resection. The imaging protocol included anatomical T2-weighted Fluid Attenuated Inversion Recovery and Fast Spin Echo images, along with 3D T1-weighted Inversion Recovery-Spoiled Gradient Recalled echo imaging pre- and post-injection of a gadolinium-based contrast agent. Diffusion-weighted imaging or DTI was obtained in the axial plane with $b = 1000$ s/mm$^2$ and either 6 gradient directions and 4 excitations or 24 gradient directions and 1 excitation or $b = 2000$ s/mm$^2$ and 55 gradient directions (echo time/repetition time = 108/1000 ms, voxel size = 1.7–2.0 × 1.7–2.0 × 2.0–3.0 mm). To calculate ADC maps, a pipeline using the FMRIB's Diffusion Toolkit was applied to the diffusion-weighted imaging and DTI data as previously described[56]. Mean tumor ADC was calculated from volumetric whole-tumor measurements. To do so, tumors were segmented on high-resolution T1 post-contrast images that were registered to ADC maps, and mean ADC values within the tumor region of interest were extracted.

## Preclinical magnetic resonance imaging

In vivo MRI was performed on a Cryogen-free 3T Bruker Biospin (Billerica) with a maximum gradient strength of 960 mT/m and a maximum slew rate of 3550 T/m/s. Multi-slice T2-weighted images were acquired using a Rapid Acquisition with Relaxation Enhancement (RARE) sequence with the following parameters: echo time/repetition time = 48/4000 ms, RARE-factor = 8, 4 signal averages, field of view = 32 × 32 mm, 25 slices with 1.0 mm slice thickness, in-plane resolution of 0.167 × 0.167 mm, which resulted in an imaging time of 6 min and 24 s.

Multi-slice diffusion tensor imaging (DTI) was acquired using a single-element diffusion-weighted echo-planar imaging sequence with the following parameters: echo time/repetition time = 30/2500 ms, 8 signal averages, 3 diffusion directions, two $b$ values per direction (b-500 and 1000 s/mm$^2$), in-plane resolution of 0.333 × 0.333 mm with a partial Fourier factor of 1.5 in the phase-encoding direction, and the same field of view, slice thickness, and slice number as the T2-weighted images. With respiratory gating, the total imaging time was 2 min and 20 s. To generate ADC maps, Horos imaging software was used to manually place polygonal regions-of-interest over the solid portion of the meningioma xenograft tumors and these volume measurements were averaged for each group. Areas of cystic change or hemorrhage as denoted by T2-weighted images were excluded.

## Code

The open-source software, tools, and packages used for data analysis in this study, as well as the version of each program, were CellRanger (v6.1.2), R (v4.2.1 and v4.3.1), Seurat R package (v4.2.1), SCTransform (0.3.5), Harmony R package (v0.1.0), Horos (v3.0), pymol (v2.x), FastQC (v0.11.9), MultiQC (v1.12), Cutadapt (v3.7), HISAT2 (v2.2.0), featureCounts (v2.0.3), DESeq2 (v1.36.0), GSEA (v.4.3.2), EnrichmentMap (v.3.3.6), Cytoscape (v.3.10.0), and AutoAnnotate (v.1.4.1). No software was used for data collection. Code is available on github (www.github.com/cdeaton380/Merlin-rescue.git).

## Statistics

No statistical methods were used to predetermine sample sizes, but our cohort sizes are similar or larger to those reported in previous publications. Data distribution was assumed to be normal, but this was not formally tested. Investigators were blinded to conditions during clinical data collection and analysis. Bioinformatic analyses were performed blind to molecular characteristics. The clinical samples used in this study were non-randomized with no intervention, and all samples were interrogated equally. Thus, controlling for covariates among clinical samples is not relevant. Unless specified otherwise, lines represent means, and error bars represent standard error of the means. Results were compared using Student's $t$ tests, Chi-squared tests, and log-rank tests, which are indicated in the text, methods, and figure legends alongside approaches used to adjust for multiple comparisons. Statistical significance is shown by $*p \leq 0.05$, $**p \leq 0.01$, or $***p \leq 0.0001$.

## Reporting summary

Further information on research design is available in the Nature Portfolio Reporting Summary linked to this article.

## Data availability

The raw single-cell RNA sequencing generated in this study have been deposited in the NCBI Gene Expression Omnibus under the accession GSE224347. Transcriptomes were simultaneously aligned against publicly available hg19 and mm10 data sets, stored on the UCSF C4 environment at refdata-cellranger-hg19-and-mm10-3.0.0/refdata-cellranger-mm10-3.0.0/. Cells with transcripts aligned to <99% of the human dataset were discarded. The raw bulk RNA sequencing data generated in this study has been deposited in the BioProject database under BioProject ID PRJNA1102120. The raw proximity labeling proteomic mass spectrometry data generated in this study have been deposited to PRIDE Proteomics Identification database under the accession 36993679 (https://www.ebi.ac.uk/pride/archive/projects/PXD053578). The remaining data are available within the Article, Supplementary Information or Source Data file. Source data are provided with this paper.

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

## Acknowledgements
The authors thank Ken Probst and Noel Sirivansanti for Illustrations and Kyla Foster for comments. This study was supported by NIH grants P50 CA097257 (DRR), R01 CA262311 (DRR), F32 CA213944 (STM), F30 CA246808 and T32 GM007618 (AC), U54 CA209891 (NJK), the Northwestern Medicine Malnati Brain Tumor Institute of the Lurie Cancer Center (STM), the UCSF Wolfe Meningioma Program Project (STM, DRR), and the Trenchard Family Charitable Fund (DRR). Sequencing reactions were performed at the UCSF Center for Advanced Technology with support from the UCSF PBBR, RRP IMIA, and NIH grant S10 OD028511.

## Author contributions
All authors made substantial contributions to the conception or design of the study; the acquisition, analysis, or interpretation of data; or drafting or revising the manuscript. All authors approved the manuscript. All authors agree to be personally accountable for individual contributions and to ensure that questions related to the accuracy or integrity of any part of the work are appropriately investigated, resolved, and the resolution documented in the literature. C.D.E., J.E.V.M., D.L.S., and D.R.R. designed the study and analyses. Experiments were performed by C.D.E., L.A., S.J.L., N.Z., Z.C., T.C.C., P.B., C.H.G.L., E.S., A.C., H.N.V., S.T.M., and D.R.R. Data analysis was performed by C.D.E., L.A., S.J.L., P.B., C.H.G.L., A.C., H.N.V., and D.R.R. The study was supervised by C.D.E., J.S.Y., N.J.K., J.E.V.M., D.L.S., and D.R.R. The manuscript was prepared by C.D.E. and D.R.R. with input from all authors.

## Competing interests
The authors declare no competing interests.
