## [Peer Review File · Nature Communications]

Merlin S13 phosphorylation controls meningioma Wnt signaling and magnetic resonance imaging featuresREVIEWER COMMENTS

Reviewer #1, expertise in sc-RNAseq for meningioma (Remarks to the Author):

This manuscript addresses the role of Merlin (encoded by NF2) in Wnt signaling in the context of meningioma. In the interest of identifying noninvasive pre-operative prognostic biomarkers, it also addresses the relationship between NF2 status, Wnt signaling, and MRI features such as ADC signal.

The NF2 gene is disrupted by mutation or deletion in roughly two out of three meningiomas, and is associated with poor prognosis. Here the authors focus on Merlin signaling in the remaining cases, in which NF2 is intact. Using a mouse model of Merlin-intact and non-intact meningioma, the authors initially observed that β -catenin and Wnt target genes are expressed more highly in Merlin-intact cells. Following this lead, they showed that in the context of meningioma, unphosphorylated Merlin is necessary for maximal signaling through the Wnt pathway, that this promotes tumor growth, and that it corresponds to low ADC signal in preoperative MRIs. The data also indicated that phosphorylated Merlin interacts with β -catenin, causing it to be sequestered at the plasma membrane and reducing signal through the Wnt pathway. In contrast, unphosphorylated Merlin does not interact with β -catenin, allowing the latter to localize to the nucleus and activate Tcf/Lef transcriptional targets, which are one canonical readout of Wnt signaling. The data in the manuscript support these claims, and the manuscript is generally well-written. However, I have some general questions about how these results relate to the bigger picture:

In Fig. 1, the survival difference between Merlin-deficient cells and Merlin-rescue cells is insignificant; how clinically important is this Wnt/Merlin/ β -catenin pathway?

How general is the role of Merlin in Wnt signaling? Is Merlin involved in Wnt signaling in nonmalignant, non-meningioma cells?

Does Merlin interact directly with β -catenin? Are there other regulators of β -catenin subcellular localization?

The scRNA-seq-derived claims could be more clearly and quantitatively presented. It is hard to see from the UMAPs in Figure 1 that there is enrichment of Merlin-rescued cells in cluster 5 – what is the statistical significance of this enrichment? In Fig. 1f there appear to be two additional clusters with higher expression of the Wnt signature – clusters 6 and 3. It would be helpful to have a clearer and more detailed explanation of the calculations underlying the data shown in Fig. 1e.

What are the implications of the data presented in Fig. 1f – in other words, that a tiny fraction of the Merlin rescue cells show elevated Wnt signature expression? How genetically uniform are these cell lines/xenografts? What underlies the variation in Wnt signature expression in these cells?

How does this data relate to other genomic and epigenetic studies of meningioma, including those by this group? Is there elevated Wnt gene expression in NF2-intact human meningiomas? Do the methylation data from Choudhuri et al (Nat. Gen. 2022) suggest any impact on Wnt signaling or Wnt targets?

The “feed-forward” nature of this control was not obvious to me – a circuit diagram showing

the elements involved in the feed-forward circuit would be helpful.

Figure 4g could be a bit more clear. It would also be helpful to include somewhere, maybe in Fig. 1, a diagram of the Wnt pathway.

Reviewer #2, expertise in meningioma (Remarks to the Author):

In this study, Eaton et al study Merlin intact meningiomas and identify important signaling pathways that drive the behavior of these tumors. They also include a cohort of 100 preoperative brain MRIs and find that better acting Merlin intact meningiomas are associated with high apparent diffusion coefficient (ADC). A similar finding was seen in animals, suggesting that ADC is inversely correlated with wnt pathway activation. The manuscript is well written and the N=100 is a large dataset, but there is a lack of detail on how the MRs were analyzed, making reproducing or comparing the results to prior or future studies challenging. In addition, while there is some controversy in the literature, high ADC has been previously shown to be associated with better behaving tumors, so this is not necessarily a novel finding (see Huang et al Neuro-Oncology 2019 for a recent review that summarizes the literature).

Major Comments:

- Stating in the abstract that they have established a noninvasive imaging biomarker is a little strongly worded. Would suggest "potential imaging biomarker".
- How was the mean ADC value generated? Meaning, did you segment the tumor on a post contrast image (high or low resolution post contrast image?) and register the ADC map to the T1 post contrast image to then extract mean ADC values within the tumor or ROI? Or did you segment the tumor on the ADC map itself and then extract the mean ADC value?
- Please confirm that the mean ADC was from the entire volume of the tumor or a selected cross section of the tumor?

Minor Comments:

- It would be interesting to report the variability of ADC within the meningiomas (suggestion only).

Reviewer #3, expertise in meningioma models (Remarks to the Author):

In this paper, the authors focused on Merlin intact meningiomas. They first try to identify the impact of inactivating/activating Merlin in meningioma cell lines (CH157MN and IOMM-Lee). Results showed that the Merlin loss did not influence histology, growth, or overall survival. Comparison of gene expression through SC RNA Seq revealed that only 1 cluster was enriched in merlin rescued xenografts: the WNT pathway effector CTNNB1/b-Catenin. Authors identified the mechanism through which Merlin regulates Wnt signaling: a feed forward mechanism that influences Merlin/b-catenin interaction and subcellular localization. Sequence analysis of the Merlin NTD (N-Terminal domain) showed that this mechanism needs Merlin S13 dephosphorylation to activate WNT Pathway. The authors try afterwards to correlate the ADC with the Wnt pathway activation in NF2 intact/Merlin Intact meningiomas.

High ADC meningioma had better outcome than low ADC meningiomas. Analysis of DNA methylation groups across ADC high versus ADC low showed that the majority of ADC high meningioma were “Merlin intact”, a DNA methylation profile previously published. Authors tried finally to correlate the ADC with Wnt pathway activation in meningioma xenograft. Overall, they conclude that Merlin post translational modifications regulate WNT signaling in tumors without NF2/Merlin inactivation, and that this mechanism is correlated to the ADC, which can be a clinical non-invasive marker of overall prognosis.

Comments:

From a general point of view, authors are working on biochemical mechanisms of Merlin intact meningiomas, which are generally grade 1 benign lesion. However, they are running all the experiments with malignant cell lines (IOMM-Lee and CH157MN). Based on previous studies, we perfectly know that these cell lines are very far from classical meningioma cells, and even more distant from classical benign meningioma cells. Moreover, Tert activation can modulate wnt/beta-catenin activity (Telomerase modulates Wnt signaling by association with target gene chromatin, Nature 2009)

This fact is underlined by the absence of prognostic significance of activating/inactivating merlin in this cell lines (which seems different from the real-life prognostic significance of the presence of NF2 alteration). Based on the many CNV and mutational alterations in these cell lines, their oncologic behavior is not mainly driven by the only NF2 status...

Maybe this fact also explains that SC RNA seq only found 1 cluster differentiating CH-157MN merlin deficient vs Merlin rescued...whereas NF2 is a main driver of meningioma tumorigenesis and transcriptional profile.

In Fig 2d, the authors showed results for CH-157 MN cells. Are their results similar on M10G cells? In Fig 2f, why are all groups treated with Wnt 3a?

In fig 3f, the authors showed that MerlinS13D rescue attenuated meningioma cell proliferation in vitro and inhibited meningioma xenograft growth in vivo (Fig. 3g) compared to MerlinWT or MerlinS13A rescue. To go further with this observation, we need to get bulk or Sc RNA seq data to compare wnt pathway (as in Fig 1) and western-blot to compare beta-catenin expression.

For the structural study of Merlin regulation, authors are using both CH157MN and M10G cell. These second cell line seems to be cultured from a meningioma. Could the authors precise its grade and mutational phenotype?

I would like to see the similar analysis using a benign Grade I meningotheial cell line, HBL-52 which is characterized by a TRAF 7 mutation with intact merlin. Could be a valuable addition to understand the role of TRAF7 mutation (passenger/driver) in the context of WT merlin.

Concerning the study on human meningiomas, I did not really catch what is the NF2/22q status in the 100 meningiomas analyzed. Were they all NF2 intact? From a theoretical point of view, it would have been logical to keep only merlin intact cases.

If there is a correlation between ADC and Wnt signaling, I am not convinced that this means a specific causality link. ADC is closely linked to the cellularity, which could be indeed increased by the pro-proliferative effect of WNT signaling... However, it could have been the same with all the pro proliferative genetic modification.

At the end of the reading, I am confused by the different experiments and the high amount of data in meningioma cells lines that does not mimic the “benign behavior” of the NF2-intact human meningiomas described by the authors.

To extend their data, the authors should analyze human NF2 -intact meningiomas. They should analyze the NF2 status, the WNT pathway activation. They should describe in this population the associated mutations (PIK3CA, AKT, TRAF7, SMO, and others) and correlate with WNT pathway. They should try to convince the reader on the role of Merlin S13 phosphorylation in the non NF2-related human meningiomas, either by Western-blot and IHC (antibody does exist- (Ma M et al, Protein cell, 2023, 14 , 137-42.

Finally, we want to know how the classical identified mutations could be only passenger mutations (the driver mechanism being through Merlin S13 phosphorylation).

For the MRI study, we would like to have more correlation with the mutational status and the wnt pathway activation. I think that the last sentence of the conclusion is an overstatement and has any clinical interest.

Can the authors comment the interest “to define meningioma biology pre-operatively using non-invasive imaging techniques” in clinical practice?

Reviewer #4, expertise in Wnt signalling, merlin and brain tumours (Remarks to the Author):

This is a paper on the potential oncogenic mechanism of merlin expressing meningiomas and extends to the very relevant question of describing imaging biomarker for molecular subtypes of meningioma. The authors, who are experts in the field, show in a series of nice complementary experiments that Merlin is in a fw mechanism regulates Wnt pathway activation. This seems to be dependent on phosphorylation of S13. The authors then show an imaging biomarker for merlin expression tumour subtype.

I have a few comments which I think need to be carefully considered before publication

Figure 1 The authors should discuss the lack of the effect of merlin expression or knockdown is because of the use of malignant cell lines

Figure 2D How do the authors explain mainly membrane and nuclear localization with Merlin S13D in figure 2D. Also are theses five different membranes/experiments? If so comparison should be done cautiously.

Have I seen figure 4c in another publication by the authors?

I think in the discussion the authors should carefully discuss how this proposed mechanism works together with e.g. know oncogenic Akt mutations in merlin expressing tumours. This is quite important. Also there is previous literature on how beta-catenin is regulated by merlin expression e.g. Zhou L et al 2011, Kim M et al 2016 which need to be discussed.

Reviewer #1, expertise in sc-RNAseq for meningioma

This manuscript addresses the role of Merlin (encoded by NF2) in Wnt signaling in the context of meningioma. In the interest of identifying noninvasive pre-operative prognostic biomarkers, it also addresses the relationship between NF2 status, Wnt signaling, and MRI features such as ADC signal. The NF2 gene is disrupted by mutation or deletion in roughly two out of three meningiomas, and is associated with poor prognosis. Here the authors focus on Merlin signaling in the remaining cases, in which NF2 is intact. Using a mouse model of Merlin-intact and non-intact meningioma, the authors initially observed that β -catenin and Wnt target genes are expressed more highly in Merlin-intact cells. Following this lead, they showed that in the context of meningioma, unphosphorylated Merlin is necessary for maximal signaling through the Wnt pathway, that this promotes tumor growth, and that it corresponds to low ADC signal in preoperative MRIs. The data also indicated that phosphorylated Merlin interacts with β -catenin, causing it to be sequestered at the plasma membrane and reducing signal through the Wnt pathway. In contrast, unphosphorylated Merlin does not interact with β -catenin, allowing the latter to localize to the nucleus and activate Tcf/Lef transcriptional targets, which are one canonical readout of Wnt signaling. The data in the manuscript support these claims, and the manuscript is generally well-written. However, I have some general questions about how these results relate to the bigger picture:

In Fig. 1, the survival difference between Merlin-deficient cells and Merlin-rescue cells is insignificant; how clinically important is this Wnt/Merlin/ β -catenin pathway?

Thank you for this suggestion. In response, we have added a paragraph to the end of our revised Discussion that articulates the limitations of our study. Therein, we state “This study should be interpreted in the context of its limitations, which include the paucity of preclinical models for studying meningioma biology. CH-157MN cells are a notoriously aggressive meningioma cell line that have been in culture for many years²², and it is possible that the lack of phenotypic differences between CH-157MN xenografts with versus without Merlin expression was due to their malignant nature (Fig. 1b, 1c). To address these limitations, we validated our results using IOMM-Lee meningioma xenografts (Extended Data Fig. 2d, e), which recapitulate the histology of human meningiomas (Extended Data Fig. 2b) better than CH-157MN xenografts (Extended Data Fig. 1a). Moreover, we validated our finding that Merlin regulates the Wnt pathway in CH-157MN meningioma xenografts (Fig. 1d-f) using M10G meningioma cells (Fig. 1g, 2c, 2e-h, 3c-e, 3i-k), which were recently derived from a non-malignant meningioma²⁷, and tested the generalizability of these results using benign HEI-193 human schwannoma cells²⁹ (Extended Data Fig. 1b, c). More broadly, robust multi-institutional data demonstrate that human Merlin-intact meningiomas have better clinical outcomes than meningiomas with bi-allelic inactivation of *NF2*^{7,8,11,12}. Thus, despite the intrinsic challenges to modeling benign meningiomas in preclinical systems, the data presented here may provide a useful platform for further interrogation in humans.” We thank to reviewer for inviting this opportunity to better contextualize our results.

With respect to the clinical significance of Wnt/Merlin/ β -catenin signaling in meningioma, we have also added a paragraph to the end of our Revised Discussion that summarizes the ways the Wnt pathway can be activated in malignant versus non-malignant meningiomas. This paragraph is presented below in the second-to-last response to Reviewer 1’s helpful suggestions for improving our manuscript.

How general is the role of Merlin in Wnt signaling? Is Merlin involved in Wnt signaling in nonmalignant, non-meningioma cells?

Our data presented in this revision show that Merlin is indeed involved in Wnt signaling in nonmalignant, non-meningioma cells. We tested this using benign HEI-193 human schwannoma cells and found that suppression of *NF2* also inhibits Wnt signaling in this context (Extended Data Fig. 1b, c). With respect to connections between Merlin and Wnt signaling in non-malignant meningioma cells themselves, M10G meningioma cells were recently derived from a non-malignant human meningioma (PMID 32968068). These data and this citation are included in our revised manuscript.

Does Merlin interact directly with β -catenin? Are there other regulators of β -catenin subcellular localization?

β -catenin subcellular localization is known to be regulated by CK1, GSK3b, JNK, b-TRcP, KDM2A and Twa1 (PMID 34980884). As described in our response to the second-to-last critique from Reviewer 1, we have incorporated these proteins and a reference to this useful review article, which contains many schematized diagrams of the Wnt pathway, in our revised Discussion.

Our proximity-labeling proteomic mass spectrometry data indicate that β -catenin is at least in proximity to Merlin, so the reviewer asks an excellent question regarding whether these proteins interact directly (or perhaps transiently or through the assistance of intermediates). To shed light on this question in our revised study, we

performed an *in vitro* binding assay using recombinant β -catenin and recombinant Merlin NTD constructs. These new data, which are presented in our revised Results and Extended Data Fig. 6b, demonstrate that, unlike the N-terminal 54 residues of TCF4 (which we used as a positive control for binding to β -catenin), the Merlin NTD is not sufficient for binding to β -catenin. In cells, Merlin forms a closed conformation in which the FERM domain interacts with the C-terminal domain, so it is possible that the NTD is necessary but not sufficient to bind to β -catenin, and that the tertiary structure of full-length Merlin is required for this interaction. Thus, in addition to incorporating these new data into our revised Results, we have also added a section to our revised Discussion that describes these considerations related to the tertiary structure of Merlin and other FERM family members. We thank the reviewer for this insightful question, and for inviting these new experiments.

The scRNA-seq-derived claims could be more clearly and quantitatively presented. It is hard to see from the UMAPs in Figure 1 that there is enrichment of Merlin-rescued cells in cluster 5 – what is the statistical significance of this enrichment? In Fig. 1f there appear to be two additional clusters with higher expression of the Wnt signature – clusters 6 and 3. It would be helpful to have a clearer and more detailed explanation of the calculations underlying the data shown in Fig. 1e.

We agree with the reviewer that the UMAP in Fig. 1d does not convey more Cluster 5 Wnt signaling meningioma cells are present in xenografts with versus without expression of *NF2*/Merlin, which is why we felt it was important to quantify the distribution of C5 Wnt signaling meningioma cells in Fig. 1e (where the difference is indeed statistically significant, $P < 0.05$, as stated in the figure legend). As is now stated in our revised Results, “Comparison of meningioma cell states across xenografts revealed only 1 cluster was enriched in Merlin rescue xenografts compared to Merlin-deficient xenografts (Fig. 1e). This cluster was distinguished by expression of the Wnt pathway effector *CTNNB1*/ β -catenin (Extended Data Fig. 3b).” Thus, the additional clusters noted in Fig. 1f do not have statistically significant differences in the level of Wnt signaling across xenograft conditions. We have provided the distribution of other meningioma cell clusters across experimental conditions in Extended Data Fig. 3a, which are not statistically significant.

What are the implications of the data presented in Fig. 1f – in other words, that a tiny fraction of the Merlin rescue cells show elevated Wnt signature expression? How genetically uniform are these cell lines/xenografts? What underlies the variation in Wnt signature expression in these cells?

Thank you for your interesting insight into our data. Our lab and others have performed sequencing of CH-157MN meningioma cells/xenografts and evidence of genetic heterogeneity has not been identified (PMID 28552950). However, our data reveal significant evidence of transcriptomic/cell-state heterogeneity *in vivo*, and we find that the meningioma cell cluster that is enriched in *CTNNB1* expression (Extended Data Fig. 3b) is also enriched in Wnt target genes (Fig. 1f) and that this cluster is increased in xenografts with versus without Merlin expression (Fig. 1e). We have revised the Results section of our manuscript to better describe these findings.

How does this data relate to other genomic and epigenetic studies of meningioma, including those by this group? Is there elevated Wnt gene expression in *NF2*-intact human meningiomas? Do the methylation data from Choudhuri et al (Nat. Gen. 2022) suggest any impact on Wnt signaling or Wnt targets?

Thank you for this excellent question. There is indeed elevated Wnt target gene expression across *NF2*-intact human meningiomas. A previous high-impact publication reported Wnt pathway activation in Merlin-intact meningiomas with *TRAF7*, *KLF4*, *PIK3CA*, *POLR2A*, or Hedgehog pathway mutations (Fig. 3c of PMID 27548314). This publication and our own previous research also suggest that the Wnt pathway can be activated in *NF2*-deficient meningiomas, possibly through epigenetic silencing of endogenous Wnt inhibitors (PMID 29590631). This putative mechanism of Wnt pathway activation in *NF2*-deficient meningiomas is of course very different from the mechanism of Merlin post-translational modification and inhibition/sequestration of β -catenin in Merlin-intact meningiomas that we propose in the current paper. Importantly, we found no evidence of epigenetic activation of the Wnt pathway in the DNA methylation data for Merlin-intact meningiomas from Choudhuri et al. (PMID 35534562).

In recognition of these complexities, we have significantly revised our Discussion to now state “Merlin-intact meningiomas encode somatic short variants (SSVs) targeting *TRAF7*, *AKT1*, or *KLF4* that may be tumor-initiating^{17,18}. Our data show Merlin^{S13} phosphorylation status and Wnt signaling modify Merlin-intact meningioma growth, but the way(s) in which this mechanism interacts with (or is influenced by) the myriad missense mutations in *TRAF7*, *AKT1*, *KLF4*, or other potential drivers that are enriched in Merlin-intact meningiomas remains to be established. Previous investigations have identified Wnt pathway activation across meningiomas with SSVs targeting *TRAF7*, *KLF4*, *PIK3CA*, *POLR2A*, the Hedgehog pathway, and even *NF2* and *SMARCB1*⁴¹ (PMID

27548314). In support of the robustness of Wnt signaling across Merlin-intact meningiomas, re-analysis of RNA sequencing data from meningiomas with SSVs targeting *TRAF7* (n=8), *PIK3CA* (n=5), *AKT1* (n=7), or *SMO* (n=14) revealed no difference in expression of Wnt target genes across genetic conditions¹³ (Supplementary Table 4). In meningiomas with adverse clinical outcomes and loss of *NF2*, the Wnt pathway may be activated by epigenetic silencing of endogenous Wnt inhibitors²⁴ (PMID 29590631), but there does not appear to be an epigenetic basis of Wnt pathway activation in meningiomas with favorable clinical outcomes and expression of *NF2*⁷ (PMID 35534562). Proximity-labeling proteomic mass spectrometry coupled with mechanistic and functional approaches demonstrate PKC and PP1A are important for this signaling mechanism in Merlin-intact meningiomas, but (1) other kinases or phosphatases, (2) other Merlin domains, or (3) other Wnt pathway members, including other mediators of β -catenin subcellular localization, such as CK1, GSK3 β , JNK, β -TRcP, KDM2A, and TWA1⁴² may also contribute to Wnt signaling in Merlin-intact meningiomas. In support of this hypothesis, we show (1) suppression of PP1A or PKC isoforms only partially regulates Merlin^{S13} phosphorylation (Fig. 3i, j), (2) epistatic Merlin^{S13D} or Merlin^{ANTD} rescue only partially restores Wnt signaling in meningioma cells (Fig. 3c, e), and (3) the non-canonical Wnt pathway regulators AMOT, AMOTL1, AMOTL2, DSG2, and DLG1 are found in proximity to Merlin alongside β -catenin in meningioma cells⁴²⁻⁴⁴ (Supplementary Table 2)."

The "feed-forward" nature of this control was not obvious to me – a circuit diagram showing the elements involved in the feed-forward circuit would be helpful. Figure 4g could be a bit more clear. It would also be helpful to include somewhere, maybe in Fig. 1, a diagram of the Wnt pathway.

We apologize that the explanations in our initially submitted manuscript were suboptimal, and we hope that the review article (PMID 34980884) mentioned in our response to the third critique from Reviewer 1 above (which is now referenced in our revised Discussion and contains many schematized diagrams of the Wnt pathway) will enhance the interpretability of our results and conclusions.

With respect to the putative feed-forward circuit we report, in unstimulated meningioma cells, we found (1) that knockdown of endogenous *NF2* negatively regulated TCF/LEF transcriptional activity (Fig. 2g, columns 1 vs 3; Fig. 2h, columns 1 vs 2), (2) that over-expression of exogenous *NF2* in cells lacking endogenous *NF2* could not rescue TCF/LEF transcriptional activity (Fig. 2h, columns 2 vs 3) and (3) that over-expression of exogenous *NF2* in cells expressing endogenous *NF2* had no impact on TCF/LEF transcriptional activity (Fig. 2g, columns 1 vs 2). However, in meningioma cells that were stimulated by Wnt3a, (4) loss of *NF2* resulted in a 50% loss of TCF/LEF transcriptional activity relative to control (Fig. 2h, columns 4 vs 5), but (5) over-expression of exogenous *NF2* either with endogenous Merlin (Fig. 2g, columns 4 vs 5) or without endogenous Merlin (Fig. 2h, columns 5 vs 6) increased the transcriptional activity of TCF/LEF transcription factors above endogenous levels. Together, these data show that Merlin supports Wnt signaling at basal levels, but when endogenous Merlin is lost, its role in Wnt signaling cannot be rescued until the Wnt pathway is back on. These data further demonstrate that the Wnt pathway needs to be on for Merlin to promote Wnt signaling, and in the context of data elsewhere in our manuscript, our results suggest that Merlin is dephosphorylated as a result of Wnt pathway activation, which in turn promotes Wnt signal transduction through β -catenin and TCF/LEF, which drive the transcription of Wnt-associated genes. In recognition of the complexity of this signaling mechanism, we have added a new paragraph to our revised Discussion (third-to-last paragraph) that summarizes these results. Moreover, we have re-presented Fig. 2g and Fig. 2h from our revised manuscript here for ease of evaluation.

sgNF2 - - + - - +
Merlin - + - - + -
Wnt3a - - - + + +

sgNF2 - + + - + +
Merlin - - + - - +
Wnt3a - - - + + +

Reviewer #2, expertise in meningioma

In this study, Eaton et al study Merlin intact meningiomas and identify important signaling pathways that drive the behavior of these tumors. They also include a cohort of 100 preoperative brain MRIs and find that better acting Merlin intact meningiomas are associated with high apparent diffusion coefficient (ADC). A similar finding was seen in animals, suggesting that ADC is inversely correlated with wnt pathway activation. The manuscript is well written and the N=100 is a large dataset, but there is a lack of detail on how the MRs were analyzed, making reproducing or comparing the results to prior or future studies challenging. In addition, while there is some controversy in the literature, high ADC has been previously shown to be associated with better behaving

tumors, so this is not necessarily a novel finding (see Huang et al Neuro-Oncology 2019 for a recent review that summarizes the literature).

We thank the reviewer for their thorough and thoughtful assessment of our study. We agree that other studies have nominated high ADC as a marker of better behaving meningiomas. We have incorporated this important consideration (and the reference suggested) into our revised Discussion. To our knowledge, our study is the first to provide mechanistic insights into the signaling pathways that underly high versus low ADC in meningiomas. Our revised Discussion now states: "Integrating data from meningioma xenografts and meningioma patients, our results establish signaling pathways that underlie meningioma ADC as a potential imaging biomarker of Wnt signaling in Merlin-intact meningiomas with S13 phosphorylation and favorable clinical outcomes (Fig. 4g). These data shed new light on how meningiomas can grow despite expression of a canonical tumor suppressor, and provide a mechanistic basis for meningioma ADC, which has been previously proposed as a non-invasive biomarker of meningioma outcomes in humans⁴⁰ (Huan et al. 2019)."

Major Comments:

- Stating in the abstract that they have established a noninvasive imaging biomarker is a little strongly worded. Would suggest "potential imaging biomarker".

We agree with the reviewer that our characterization of ADC (especially in the context of the previously published literature noted above) was too strong. As suggested, our revised Abstract now states "To translate these findings to clinical practice, we define mechanisms underlying a potential imaging biomarker that could be used to guide treatment de-escalation or imaging surveillance for patients with favorable meningiomas." As illustrated in the related revisions to our Discussion described above, we have also reframed the conceptual advance of our study away from discovering the value of ADC itself (which we did not mean to convey) and toward unraveling the signaling mechanisms underlying meningioma ADC. We thank the reviewer for bringing this to our attention.

- How was the mean ADC value generated? Meaning, did you segment the tumor on a post contrast image (high or low resolution post contrast image?) and register the ADC map to the T1 post contrast image to then extract mean ADC values within the tumor or ROI? Or did you segment the tumor on the ADC map itself and then extract the mean ADC value?

Tumor were segmented on high resolution T1 post contrast images that were registered to ADC maps, and mean ADC values within the tumor region of interest were extracted. We have incorporated these details of this approach into our revised Methods, and we thank the reviewer for the opportunity to clarify this aspect of our study.

- Please confirm that the mean ADC was from the entire volume of the tumor or a selected cross section of the tumor?

We confirm that mean ADC was calculated from volumetric whole-tumor measurements (not selected cross sections) and have incorporated these details of our approach into our revised Methods.

Minor Comments:

- It would be interesting to report the variability of ADC within the meningiomas (suggestion only).

Thank you for this suggestion, which we have explored in a previous publication specifically investigating meningioma intratumor molecular heterogeneity in the context of meningioma intratumor ADC heterogeneity (PMID 32968068). Our previous investigations revealed that ADC variability was increased in high versus low grade meningiomas (Fig. 3b of PMID 32968068), and that intratumor heterogeneity correlated with intratumor Wnt signaling (Fig. 3d of PMID 32968068). We have incorporated this reference and a note about the potential impact of intratumor heterogeneity on meningioma ADC measurements in the concluding paragraph of our revised Discussion. There, we state "With respect to translation to human patients, intratumor heterogeneity can influence meningioma ADC measurements and correlates with intratumor Wnt signaling²⁷. It is unknown if intratumor heterogeneity will be a barrier to adoption of non-invasive imaging biomarkers of meningioma biology, but emerging evidence suggests that intratumor heterogeneity is an important determinant of meningioma susceptibility to molecular therapy⁵⁵. It is likely that multiple molecular mechanisms contribute to meningioma ADC, and future clinical trials that incorporate imaging and biomarker studies will be necessary to answer these questions. In the interim, our results identify a signaling pathway that underlies a potential imaging biomarker that could be used to guide treatment de-escalation or imaging surveillance for patients with favorable meningiomas."

Reviewer #3, expertise in meningioma models

In this paper, the authors focused on Merlin intact meningiomas. They first try to identify the impact of inactivating/activating Merlin in meningioma cell lines (CH157MN and IOMM-Lee). Results showed that the Merlin loss did not influence histology, growth, or overall survival. Comparison of gene expression through SC RNA Seq revealed that only 1 cluster was enriched in merlin rescued xenografts: the WNT pathway effector CTNNB1/b-Catenin. Authors identified the mechanism through which Merlin regulates Wnt signaling: a feed forward mechanism that influences Merlin/b-catenin interaction and subcellular localization. Sequence analysis of the Merlin NTD (N-Terminal domain) showed that this mechanism needs Merlin S13 dephosphorylation to activate WNT Pathway. The authors try afterwards to correlate the ADC with the Wnt pathway activation in NF2 intact/Merlin Intact meningiomas. High ADC meningioma had better outcome than low ADC meningiomas. Analysis of DNA methylation groups across ADC high versus ADC low showed that the majority of ADC high meningioma were “Merlin intact”, a DNA methylation profile previously published. Authors tried finally to correlated the ADC with Wnt pathway activation in meningioma xenograft. Overall, they conclude that Merlin post translational modifications regulate WNT signaling in tumors without NF2/Merlin inactivation, and that this mechanism is correlated to the ADC, which can be a clinical non-invasive marker of overall prognosis.

We thank the reviewer for their thorough and thoughtful review of our study.

Comments:

From a general point of view, authors are working on biochemical mechanisms of Merlin intact meningiomas, which are generally grade 1 benign lesion. However, they are running all the experiments with malignant cell lines (IOMM-Lee and CH157MN). Based on previous studies, we perfectly know that these cell lines are very far from classical meningioma cells, and even more distant from classical benign meningioma cells. Moreover, Tert activation can modulate wnt/beta-catenin activity (Telomerase modulates Wnt signaling by association with target gene chromatin, Nature 2009)

Previous investigations demonstrate that approximately 20% of Merlin-intact meningiomas are WHO grade 2 or WHO grade 3 (Fig. 1d of PMID 35534562). However, the reviewer's point is well taken that IOMM-Lee and CH-157MN meningioma cells are notoriously aggressive. We used these cells for some (but not all) of our studies because these cells can be grown as xenografts *in vivo* (unlikely benign meningioma cell lines). To address these limitations, we validated our results using M10G meningioma cells, which are not TERT immortalized (PMID 32968068), and benign HEI-193 schwannoma cells. To better contextualize our findings in the context of available preclinical models, we have added a paragraph to the end of our revised Discussion that articulates the limitations of our study. Therein, we state “This study should be interpreted in the context of its limitations, which include the paucity of preclinical models for studying meningioma biology. CH-157MN cells are a notoriously aggressive meningioma cell line that have been in culture for many years²², and it is possible that the lack of phenotypic differences between CH-157MN xenografts with versus without Merlin expression was due to the malignant nature of these cells (Fig. 1b, 1c). To address this limitation, we validated our results using IOMM-Lee meningioma xenografts (Extended Data Fig. 2d, e), which recapitulate the histology of human meningiomas (Extended Data Fig. 2b) better than CH-157MN xenografts (Extended Data Fig. 1a). Moreover, we validated our finding that Merlin regulates the Wnt pathway in CH-157MN meningioma xenografts (Fig. 1d-f) using M10G meningioma cells (Fig. 1g, 2c, 2e-h, 3c-e, 3i-k), which were recently derived from a non-malignant meningioma²⁷, and tested the generalizability of these results using benign HEI-193 human schwannoma cells²⁹ (Extended Data Fig. 1b, c). More broadly, robust multi-institutional data demonstrate that human Merlin-intact meningiomas have better clinical outcomes than meningiomas with bi-allelic inactivation of *NF2*^{7,8,11,12}. Thus, despite the intrinsic challenges to modeling benign meningiomas in preclinical systems, the data presented here may provide a useful platform for further interrogation in humans.” We thank to reviewer for inviting this opportunity to better contextualize our results.

This fact is underlined by the absence of prognostic significance of activating/inactivating merlin in this cell lines (which seems different from the real-life prognostic significance of the presence of NF2 alteration). Based on the many CNV and mutational alterations in these cell lines, their oncologic behavior is not mainly driven by the only NF2 status...

We agree with the reviewer that meningioma drivers can be multifactorial, even in low-grade meningiomas which are often Merlin-intact and enriched in somatic short variants targeting *TRAF7*, *PIK3CA*, *AKT1*, *KLF4*, or the Hedgehog pathway. The objective of this study was to understand what Merlin expression or loss contributes

across meningiomas, which is why it was important to test our hypotheses across a diversity of cell and xenograft models. To better frame this objective, our revised Introduction now states “Here we test the hypothesis that understanding signaling mechanisms associated with *NF2*/Merlin itself may shed light on meningioma biology and elucidate strategies to define meningioma biology pre-operatively using non-invasive imaging techniques.”

Maybe this fact also explains that SC RNA seq only found 1 cluster differentiating CH-157MN merlin deficient vs Merlin rescued...whereas *NF2* is a main driver of meningioma tumorigenesis and transcriptional profile. In Fig 2d, the authors showed results for CH-157 MN cells. Are their results similar on M10G cells? In Fig 2f, why are all groups treated with Wnt 3a?

Our results in Fig. 2d in CH-157MN cells are indeed similar in M10G cells (please see Extended Data Fig. 5a), where we see cancer-associated Merlin missense mutants have significantly different subcellular localizations compared to wildtype Merlin, but Merlin S13 unphosphorylatable and phospho-mimetic mutants do not have significantly different subcellular localizations compared to wildtype Merlin. In Fig. 2f we wanted to understand if Merlin was required for maximal Wnt pathway activation in meningioma cells even after β -catenin overexpression, which is why Wnt3a stimulation was included in all conditions. These data support our hypothesis that Merlin contributes to a feed-forward mechanism of Wnt pathway activation, that we now describe in more granular detail in our revised Discussion. We show other experiments with versus without Wnt3a stimulation in Fig. 1g, 2e, 2g, and 2h. To better contextualize these results, our revised Results section now states “ β -catenin suppression using siRNAs (si*CTNNB1*) inhibited meningioma Wnt signaling (Fig. 2e and Extended Data Fig. 5c), but Merlin was required for maximal Wnt pathway activation in meningioma cells even after β -catenin overexpression (Fig. 2f and Extended Data Fig. 5d).” The relevant section of our revised Discussion now states: “With respect to the putative feed-forward circuit we report, in unstimulated meningioma cells, we found (1) that knockdown of endogenous *NF2* negatively regulated TCF/LEF transcription activity (Fig. 2g, columns 1 vs 3, and Fig. 2h, columns 1 vs 2), (2) that over-expression of exogenous *NF2* in cells lacking endogenous *NF2* could not rescue TCF/LEF transcriptional activity (Fig. 2h, columns 2 vs 3) and (3) that over-expression of exogenous *NF2* in cells expressing endogenous *NF2* had no impact on TCF/LEF transcriptional activity (Fig. 2g, columns 1 vs 2). However, after meningioma cell stimulation with Wnt3a, (4) knockdown of endogenous *NF2* resulted in a 50% loss of TCF/LEF transcriptional activity (Fig. 2h, columns 4 vs 5), but (5) over-expression of exogenous *NF2* either with endogenous Merlin (Fig. 2g, columns 4 vs 5) or without endogenous Merlin (Fig. 2h, columns 5 vs 6) increased TCF/LEF transcriptional activity above endogenous levels. Together, these data show that Merlin supports Wnt signaling at basal levels, but when endogenous Merlin is lost, its role in Wnt signaling cannot be rescued until the Wnt pathway is stimulated. These data further demonstrate that the Wnt pathway needs to be on for Merlin to promote Wnt signaling, and in the context of the other data in our study, our results suggest that Merlin is dephosphorylated as a result of Wnt pathway activation, which in turn promotes Wnt signal transduction through β -catenin and TCF/LEF.”

In fig 3f, the authors showed that MerlinS13D rescue attenuated meningioma cell proliferation in vitro and inhibited meningioma xenograft growth in vivo (Fig. 3g) compared to MerlinWT or MerlinS13A rescue. To go further with this observation, we need to get bulk or Sc RNA seq data to compare wnt pathway (as in Fig 1) and western-blot to compare beta-catenin expression.

As the reviewer noted above with respect to Fig. 1, differences in single-cell RNA sequencing of CH-157MN meningioma xenografts with versus without Merlin rescue are subtle, and it is unlikely that further single-cell RNA sequencing of a xenograft model that has limitations (as the reviewer also astutely noted above) will provide significant conceptual advances. To determine if other approaches could be used to generate more robust results, we performed bulk RNA sequencing on CH-157MN meningioma xenografts with (n=4) versus without (n=3) Merlin rescue, and we were unable to identify gene expression changes that enhance understanding of our single-cell RNA sequencing results. Broadly, these data underscore the importance of using single-cell RNA sequencing, as we do in Fig. 1, to examine subtle phenotypes, such as Wnt signaling in meningioma xenograft models. These new data are presented in Extended Data Fig. 6f, Extended Data Fig. 6g, and Supplementary Table 3 of our revised manuscript. To go a step further, we also performed QPCR to assess for Wnt target gene expression with rescue of Merlin versus Merlin^{S13A} versus Merlin^{S13D}. These new data, which are presented in Extended Data Fig. 6c of our revised manuscript, show enrichment of the Wnt target genes *AXIN2* and *DKK1* with Merlin^{S13A} rescue. These data broadly support our hypothesis that Merlin phosphorylation at S13 inhibits the Wnt pathway in Merlin-intact meningiomas, and we thank the reviewer for the opportunity to enhance the robustness of our results.

For the structural study of Merlin regulation, authors are using both CH157MN and M10G cell. These second cell line seems to be cultured from a meningioma. Could the authors precise its grade and mutational phenotype?

M10G cells were indeed recently cultured from a meningioma, WHO grade 2, with copy number variants deleting 1 copy of chromosome 1p and 1 copy of chromosome 22q but without somatic short variants in the remaining copy of *NF2* or other potential oncogenic drivers (PMID 32968068). Thus, this cell line retains a biochemically functional copy of Merlin. This cell line grows slowly and does not form xenografts *in vivo*, which is why it was necessary to incorporate other meningioma cell lines that do grow *in vivo*. We have incorporated this reference and a description of the molecular architecture of M10G cells into the Results of our revised manuscript.

I would like to see the similar analysis using a benign Grade I meningothelial cell line, HBL-52 which is characterized by a TRAF 7 mutation with intact merlin. Could be a valuable addition to understand the role of TRAF7 mutation (passenger/driver) in the context of WT merlin.

Our lab has worked with HBL-52 cells in the past (PMID 29590631), and we have been unable to stably engineer these cells to express CRISPRi machinery or sgRNAs, or to suppress gene function using orthogonal genetic techniques. Consistent with this previous experience, we attempted to suppress *NF2*/Merlin in these cells for this revision, and were unfortunately unsuccessful. However, a previous high-impact publication reported Wnt pathway activation in meningiomas with *TRAF7* mutations and in meningiomas with other somatic short variants that are enriched in Merlin-intact tumors (Fig. 3c of PMID 27548314). This publication and own previous research also suggest that the Wnt pathway can be activated in *NF2*-deficient meningiomas, possibly through epigenetic silencing of endogenous Wnt inhibitors (PMID 29590631). This putative mechanism of Wnt pathway activation in *NF2*-deficient meningiomas is of course very different from the mechanism of Merlin post-translational modification and inhibition/sequestration of β -catenin in Merlin-intact meningiomas that we propose in the current paper. In recognition of these complexities, we have significantly revised our Discussion to now state “Merlin-intact meningiomas encode SSVs targeting *TRAF7*, *AKT1*, or *KLF4* that may be tumor-initiating^{17,18}. Our data show Merlin^{S13} phosphorylation status and Wnt signaling modify Merlin-intact meningioma growth, but the way(s) in which this mechanism interacts with (or is influenced by) the myriad missense mutations in *TRAF7*, *AKT1*, *KLF4*, or other potential driver mutations that are enriched in Merlin-intact meningiomas remains to be established. Previous investigations have identified Wnt pathway activation across meningiomas with SSVs targeting *TRAF7*, *KLF4*, *PIK3CA*, *POLR2A*, the Hedgehog pathway, and even *NF2* and *SMARCB1*⁴¹. In support of the robustness of Wnt signaling across Merlin-intact meningiomas, re-analysis of RNA sequencing data from meningiomas with SSVs targeting *TRAF7* (n=8), *PIK3CA* (n=5), *AKT1* (n=7), or *SMO* (n=14) revealed no difference in expression of Wnt target genes across genetic conditions¹³ (Supplementary Table 4). In meningiomas with adverse clinical outcomes and loss of *NF2*, the Wnt pathway may be activated by epigenetic silencing of endogenous Wnt inhibitors²⁴, but there does not appear to be an epigenetic basis of Wnt pathway activation in meningiomas with favorable clinical outcomes and expression of *NF2*⁷. Proximity-labeling proteomic mass spectrometry coupled with mechanistic and functional approaches demonstrate PKC and PP1A are important for this signaling mechanism in Merlin-intact meningiomas, but (1) other kinases or phosphatases, (2) other Merlin domains, or (3) other Wnt pathway members, including other mediators of β -catenin subcellular localization, such as CK1, GSK3 β , JNK, β -TRcP, KDM2A, and TWA1⁴² may also contribute to Wnt signaling in Merlin-intact meningiomas. In support of this hypothesis, we show (1) suppression of PP1A or PKC isoforms only partially regulates Merlin^{S13} phosphorylation (Fig. 3i, j), (2) epistatic Merlin^{S13D} or Merlin^{ΔNTD} rescue only partially restores Wnt signaling in meningioma cells (Fig. 3c, e), and (3) the non-canonical Wnt pathway regulators AMOT, AMOTL1, AMOTL2, DSG2, and DLG1 are found in proximity to Merlin alongside β -catenin in meningioma cells^{43–45} (Supplementary Table 2).”

Concerning the study on human meningiomas, I did not really catch what is the *NF2*/*22q* status in the 100 meningiomas analyzed. Were they all *NF2* intact? From a theoretical point of view, it would have been logical to keep only merlin intact cases.

The molecular groups of the meningiomas analyzed for clinical outcomes in the context of whole-tumor ADC is shown in Fig. 4c, which demonstrates that a majority of tumors with high ADC values are indeed Merlin-intact. Genomic analyses of these tumors were previously published (PMID 35534562 and 37944590) but were not previously analyzed in the context of magnetic resonance imaging features.

If there is a correlation between ADC and Wnt signaling, I am not convinced that this means a specific causality link. ADC is closely linked to the cellularity, which could be indeed increased by the pro-proliferative effect of WNT signaling... However, it could have been the same with all the pro proliferative genetic modification.

We agree that multiple mechanisms may contribute to meningioma ADC and have revised our Discussion to articulate this hypothesis. We present these considerations (which are provided below for ease of evaluation) in the context of related considerations of intratumor heterogeneity, which were invited by Reviewer 4.

“With respect to translation to human patients, intratumor heterogeneity can influence meningioma ADC measurements and correlates with intratumor Wnt signaling²⁷. It is unknown if intratumor heterogeneity will be a barrier to adoption of non-invasive imaging biomarkers of meningioma biology, but emerging evidence suggests that intratumor heterogeneity is an important determinant of meningioma susceptibility to molecular therapy⁵⁵. It is likely that multiple molecular mechanisms contribute to meningioma ADC, and future clinical trials that incorporate imaging and biomarker studies will be necessary to answer these questions. In the interim, our results identify a signaling pathway that underlies a potential imaging biomarker that could be used to guide treatment de-escalation or imaging surveillance for patients with favorable meningiomas.”

At the end of the reading, I am confused by the different experiments and the high amount of data in meningioma cells lines that does not mimic the “benign behavior” of the NF2-intact human meningiomas described by the authors. To extend their data, the authors should analyze human NF2 -intact meningiomas. They should analyze the NF2 status, the WNT pathway activation. They should describe in this population the associated mutations (PIK3CA, AKT, TRAF7, SMO, and others) and correlate with WNT pathway. They should try to convince the reader on the role of Merlin S13 phosphorylation in the non NF2-related human meningiomas, either by Western-blot and IHC (antibody does exist- (Ma M et al, Protein cell, 2023, 14 , 137-42).

We thank the reviewer for these suggestions and note that many of our analyses were performed in non-malignant M10G meningioma cells or benign HEI-193 schwannoma cells (as articulated in response to the first critique from Reviewer 3 above). We also thank the reviewer for drawing our attention to this publication reporting alternative kinases that phosphorylate Merlin S13 (it was encouraging to see PRKCA among the candidates identified in Fig. 1i of this paper). We have incorporated a reference to this publication in our revised Discussion, but it does not appear that IHC with Merlin^{PS13} antibodies or immunoblots with Merlin^{PS13} antibodies were performed across heterogeneous tissues (such as tumors) in this publication. For our revision, we attempted to use the novel phospho-specific antibody recognizing Merlin^{PS13} that we generated (Fig. 3i, j, Extended Data Fig. 6e) to perform immunoblot and IHC analyses of human meningioma, but our custom antibody is limited by its sensitivity to only overexpressed Merlin and results were likely confounded by cell type heterogeneity.

With respect to correlating Wnt signaling across human NF2-intact meningiomas, we have now re-analyzed bulk RNA sequencing data across the conditions requested and present these new data in the context of the published literature on this topic. These new experimental results and prose are provided below for ease of evaluation:

“Previous investigations have identified Wnt pathway activation across meningiomas with SSVs targeting *TRAF7*, *KLF4*, *PIK3CA*, *POLR2A*, the Hedgehog pathway, and even *NF2* and *SMARCB1*⁴¹. In support of the robustness of Wnt signaling across Merlin-intact meningiomas, re-analysis of RNA sequencing data from meningiomas with SSVs targeting *TRAF7* (n=8), *PIK3CA* (n=5), *AKT1* (n=7), or *SMO* (n=14) revealed no difference in expression of Wnt target genes across genetic conditions¹³ (Supplementary Table 4).”

Finally, we want to know how the classical identified mutations could be only passenger mutations (the driver mechanism being through Merlin S13 phosphorylation).

We certainly do not contend that all classically identified mutations in meningiomas with favorable clinical outcomes (*TRAF7*, *AKT1*, *KLF4*, etc) are merely passengers. However, some of these mutations do not appear to drive meningioma tumorigenesis in mice, such as *PIK3CA* and *AKT* (PMID 34496175), and other mutations targeting the Hedgehog pathway do not appear to drive meningioma cell proliferation or susceptibility to Hedgehog pathway inhibitors *in vitro* (PMID 32690089 and 36382111). We cite these papers, which provide a rationale for studying Merlin itself in NF2-intact meningiomas, in the Introduction and Discussion of our revised manuscript. In our revised Introduction, we now state “Merlin-intact meningiomas can encode somatic short variants (SSVs) targeting *TRAF7*, *PIK3CA*, *AKT1*, *KLF4*, or the Hedgehog pathway^{17,18}, but some of these variants do not drive meningioma tumorigenesis in mice, and others do not drive meningioma cell proliferation

or susceptibility to molecular therapy *in vitro*^{19–21}. These data suggest that some SSVs in Merlin-intact meningiomas may be passengers that do not influence meningioma tumorigenesis or perhaps modify (rather than drive) meningioma biology. More broadly, these data indicate that biochemical mechanisms driving Merlin-intact meningiomas are incompletely understood.”

For the MRI study, we would like to have more correlation with the mutational status and the wnt pathway activation. I think that the last sentence of the conclusion is an overstatement and has any clinical interest. Can the authors comment the interest “to define meningioma biology pre-operatively using non-invasive imaging techniques” in clinical practice?

With respect to our analyses of meningioma ADC in the context of meningioma molecular features, our revised discussion now states “To study associations between meningioma ADC and tumor biology, 100 preoperative MRIs from meningiomas with available DNA methylation profiling and targeted exome sequencing of the *NF2* locus were retrospectively reviewed and imaging features were analyzed in the context of clinical follow-up data^{7,13}.” We regret that sequencing data for the myriad SSVs that are enriched in Merlin-intact meningiomas (although each to only a small degree) is not available for these tumors. Moreover, as previous high-impact publications have already demonstrated Wnt pathway activation across meningiomas with *TRAF7*, *KLF4*, *PIK3CA*, *POLR2A*, and the Hedgehog pathway (PMID 27548314), our interpretation is that further sequencing analyses would be redundant with the published literature. The conceptual advance offered by our study is the identification of consensus biochemical mechanism of Wnt pathway activation in Merlin-intact meningiomas, which (from the published literature and our own re-analyses my RNA sequencing data described above) appears to be conserved across the DNA mutations that are found in these tumors.

We agree with the reviewer’s intimation that our characterization of ADC as a ready-to-use non-invasive imaging biomarker was perhaps too strong. As suggested, our revised Abstract now states “To translate these findings to clinical practice, we define mechanisms underlying a potential imaging biomarker that could be used to guide treatment de-escalation or imaging surveillance for patients with favorable meningiomas.”

Reviewer #4, expertise in Wnt signalling, merlin and brain tumours

This is a paper on the potential oncogenic mechanism of merlin expressing meningiomas and extends to the very relevant question of describing imaging biomarker for molecular subtypes of meningioma. The authors, who are experts in the field, show in a series of nice complementary experiments that Merlin is in a fw mechanism regulates Wnt pathway activation. This seems to be dependent on phosphorylation of S13. The authors then show an imaging biomarker for merlin expression tumour subtype. I have a few comments which I think need to be carefully considered before publication

Thank you for your thoughtful suggestions to improve our study.

Figure 1 The authors should discuss the lack of the effect of merlin expression or knockdown is because of the use of malignant cell lines

Thank you for this suggestion. In response, we have added a paragraph to the end of our revised Discussion that articulates the limitations of our study. Therein, we state “This study should be interpreted in the context of its limitations, which include the paucity of preclinical models for studying meningioma biology. CH-157MN cells are a notoriously aggressive meningioma cell line that have been in culture for many years²², and it is possible that the lack of phenotypic differences between CH-157MN xenografts with versus without Merlin expression was due to their malignant nature (Fig. 1b, 1c). To address these limitations, we validated our results using IOMM-Lee meningioma xenografts (Extended Data Fig. 2d, e), which recapitulate the histology of human meningiomas (Extended Data Fig. 2b) better than CH-157MN xenografts (Extended Data Fig. 1a). Moreover, we validated our finding that Merlin regulates the Wnt pathway in CH-157MN meningioma xenografts (Fig. 1d-f) using M10G meningioma cells (Fig. 1g, 2c, 2e-h, 3c-e, 3i-k), which were recently derived from a non-malignant meningioma²⁷, and tested the generalizability of these results using benign HEI-193 human schwannoma cells²⁹ (Extended Data Fig. 1b, c).” We thank to reviewer for inviting this opportunity to better contextualize our results.

Figure 2D How do the authors explain mainly membrane and nuclear localization with Merlin S13D in figure 2D. Also are theses five different membranes/experiments? If so comparison should be done cautiously. Have I seen figure 4c in another publication by the authors?

These immunoblots are indeed from 5 different membranes, but they were run at the same time from the same integrated experiment with equal protein loading and equal exposure times. We have now clarified these important aspects of our approach in our revised Methods section. We entirely agree with the reviewer that

comparison across experimental conditions should be done with caution in scenarios such as thus. Thus, qualitatively comparing within conditions in this experiment, the data in Fig. 2d demonstrate that the subcellular localization of Merlin is similar with unphosphorylatable versus phospho-mimetic mutations at S13 (with enrichment at the membrane and cytoplasm, and perhaps mild enrichment of Merlin^{S13D} in the nucleus compared to Merlin^{S13A}, but we hesitant to make such comparisons across conditions). Despite these similar biochemical profiles, which are supported by immunofluorescence data in Fig. 2a, there is notable enrichment of β -catenin at the membrane with Merlin^{S13D} rescue in Fig. 2d. These results contrast with the subcellular localization of cancer-associated missense Merlin constructs (L46R, A211D) in Fig. 2d, which is heterogeneous. To present these results in an appropriately cautious light, our revised Results now states: “Rescue of Merlin^{S13D} but not Merlin^{S13A} sequestered β -catenin at the plasma membrane in meningioma cells without causing dramatic differences in the subcellular localization of Merlin itself (Fig. 2d).” We thank the reviewer for their careful evaluation of our data, and for the opportunity to explain these results.

We have not previously published the imaging data in Fig. 4c, although we have used this style of stacked bar plot to display clinical metadata across meningioma DNA methylation groups in prior publications (PMID 36227281 and 35534562) and in many prior scientific presentations. Our last studies on meningioma imaging (PMID 31608329 and 32968068) were published before our investigations of meningioma DNA methylation groups (PMID 36227281 and 35534562). Thus, the current study by Eaton et al. is the first integrated analysis of meningioma imaging and molecular features from our group.

I think in the discussion the authors should carefully discuss how this proposed mechanism works together with e.g. know oncogenic Akt mutations in merlin expressing tumours. This is quite important. Also there is previous literature on how beta-catenin is regulated by merlin expression e.g. Zhou L et al 2011, Kim M et al 2016 which need to be discussed.

Thank you for inviting these improvements. In responses, we have incorporated these references and the topics suggested into our revised Discussion, which has undergone a major revision. The relevant sections of our revised Discussion are provided below for ease of evaluation.

“Our finding that Merlin post-translational modification can promote Wnt signaling is unexpected considering the well-described tumor suppressor functions of *NF2*. CNVs deleting *NF2* on chromosome 22q are early events underlying Immune-enriched or Hypermitotic meningioma tumorigenesis^{7,27}, but Merlin-intact meningiomas encode SSVs targeting *TRAF7*, *AKT1*, or *KLF4* that may be tumor-initiating^{17,18}. Our data show Merlin^{S13} phosphorylation status and Wnt signaling modify Merlin-intact meningioma growth, but the way(s) in which this mechanism interacts with (or is influenced by) the myriad missense mutations in *TRAF7*, *AKT1*, *KLF4*, or other potential driver mutations that are enriched in Merlin-intact meningiomas remains to be established. Previous investigations have identified Wnt pathway activation across meningiomas with SSVs targeting *TRAF7*, *KLF4*, *PIK3CA*, *POLR2A*, the Hedgehog pathway, and even *NF2* and *SMARCB1*⁴¹. In support of the robustness of Wnt signaling across Merlin-intact meningiomas, re-analysis of RNA sequencing data from meningiomas with SSVs targeting *TRAF7* (n=8), *PIK3CA* (n=5), *AKT1* (n=7), or *SMO* (n=14) revealed no difference in expression of Wnt target genes across genetic conditions¹³ (Supplementary Table 4). In meningiomas with adverse clinical outcomes and loss of *NF2*, the Wnt pathway may be activated by epigenetic silencing of endogenous Wnt inhibitors²⁴, but there does not appear to be an epigenetic basis of Wnt pathway activation in meningiomas with favorable clinical outcomes and expression of *NF2*⁷. Proximity-labeling proteomic mass spectrometry coupled with mechanistic and functional approaches demonstrate PKC and PP1A are important for this signaling mechanism in Merlin-intact meningiomas, but (1) other kinases or phosphatases, (2) other Merlin domains, or (3) other Wnt pathway members, including other mediators of β -catenin subcellular localization, such as CK1, GSK3 β , JNK, β -TRCp, KDM2A, and TWA1⁴² may also contribute to Wnt signaling in Merlin-intact meningiomas. In support of this hypothesis, we show (1) suppression of PP1A or PKC isoforms only partially regulates Merlin^{S13} phosphorylation (Fig. 3i, j), (2) epistatic Merlin^{S13D} or Merlin^{ANTD} rescue only partially restores Wnt signaling in meningioma cells (Fig. 3c, e), and (3) the non-canonical Wnt pathway regulators AMOT, AMOTL1, AMOTL2, DSG2, and DLG1 are found in proximity to Merlin alongside β -catenin in meningioma cells^{43–45} (Supplementary Table 2).

Crosstalk between Merlin and the Wnt pathway is complex, and interactions between Merlin and LRP6 can inhibit the Wnt pathway in non-cancer cell lines, schwannoma cells, tissues from patients with neurofibromatosis type 2, and xenopus embryos⁴⁶. Loss of Merlin is associated with β -catenin phosphorylation and Wnt signaling in schwannoma cells⁴⁷, and in meningiomas, the Wnt pathway can be activated by multiple mechanisms in tumors with unfavorable clinical outcomes²⁴. Thus, our results may also shed light on potential targets for future

molecular therapies that could be used to treat molecularly-defined groups of meningiomas or schwannomas that are resistant to standard interventions. With respect to the putative feed-forward circuit we report, in unstimulated meningioma cells, we found (1) that knockdown of endogenous *NF2* negatively regulated TCF/LEF transcription activity (Fig. 2g, columns 1 vs 3, and Fig. 2h, columns 1 vs 2), (2) that over-expression of exogenous *NF2* in cells lacking endogenous *NF2* could not rescue TCF/LEF transcriptional activity (Fig. 2h, columns 2 vs 3) and (3) that over-expression of exogenous *NF2* in cells expressing endogenous *NF2* had no impact on TCF/LEF transcriptional activity (Fig. 2g, columns 1 vs 2). However, after meningioma cell stimulation with Wnt3a, (4) knockdown of endogenous *NF2* resulted in a 50% loss of TCF/LEF transcriptional activity (Fig. 2h, columns 4 vs 5), but (5) over-expression of exogenous *NF2* either with endogenous Merlin (Fig. 2g, columns 4 vs 5) or without endogenous Merlin (Fig. 2h, columns 5 vs 6) increased TCF/LEF transcriptional activity above endogenous levels. Together, these data show that Merlin supports Wnt signaling at basal levels, but when endogenous Merlin is lost, its role in Wnt signaling cannot be rescued until the Wnt pathway is stimulated. These data further demonstrate that the Wnt pathway needs to be on for Merlin to promote Wnt signaling, and in the context of the other data in our study, our results suggest that Merlin is dephosphorylated as a result of Wnt pathway activation, which in turn promotes Wnt signal transduction through β -catenin and TCF/LEF.”

REVIEWERS' COMMENTS

Reviewer #1 (Remarks to the Author):

The authors have addressed all of my concerns very thoroughly.

Reviewer #2 (Remarks to the Author):

The authors answered all my comments.

Reviewer #3 (Remarks to the Author):

The reviewers responded to my questions, at the very least, they show that they can't use benign tumor cells, which is a pity.
nothing to add to their very long responses.

Reviewer #4 (Remarks to the Author):

The authors thoroughly revised the manuscript. However I am still not convinced the lack of the effect of merlin expression or knockdown is because of the use of malignant cell lines. Just adding another malignant cell line IOMM-Lee does not answer the question. I think this is an important issue as it about relevance of the mechanism the authors show (also in lower grade cell lines)

REVIEWERS' COMMENTS

Reviewer #1 (Remarks to the Author):

The authors have addressed all of my concerns very thoroughly.

We thank the reviewer for their kind response.

Reviewer #2 (Remarks to the Author):

The authors answered all my comments.

We are grateful to the reviewer that we were able to address your comments.

Reviewer #3 (Remarks to the Author):

The reviewers responded to my questions, at the very least, they show that they can't use benign tumor cells, which is a pity.
nothing to add to their very long responses.

We agree with the reviewer that the lack of models of benign meningioma is a problem in the meningioma field. Our lab is working on creating better model systems and we hope to use these in our future research projects.

Reviewer #4 (Remarks to the Author):

The authors thoroughly revised the manuscript. However I am still not convinced the lack of the effect of merlin expression or knockdown is because of the use of malignant cell lines. Just adding another malignant cell line IOMM-Lee does not answer the question. I think this is an important issue as it about relevance of the mechanism the authors show (also in lower grade cell lines)

We thank the reviewer for their thoughtful comments. Unfortunately, there is a lack of representative benign meningioma cell lines that proliferate and transfect well in cell culture, or form xenograft tumours. We sympathise with this reviewer and recognize that it is a pressing issue in the field, so we are working towards generating better models for use in future projects. In the interim, our manuscript incorporates M10G meningioma cells, which are not malignant and were derived from a lower grade tumor.